# Study on Genomic Diversity, Prophage Distribution of Bovine-Derived *Staphylococcus aureus* and Their Association with Antimicrobial Resistance

**DOI:** 10.3390/microorganisms13122723

**Published:** 2025-11-28

**Authors:** Yaqian Liang, Wenjing Wang, Yuling Guo, Meihui Tian, Junkai Wang, Haihong Hao

**Affiliations:** 1College of Animal Science and Technology, Shihezi University, Shihezi 832000, China; 13134600046@163.com (Y.L.); m18331120516@163.com (W.W.); 15039717260@163.com (Y.G.); 15299963610@163.com (M.T.); aaakumainmymind@outlook.com (J.W.); 2National Key Laboratory of Agricultural Microbiology Resources Exploration and Utilization, Huazhong Agricultural University, Wuhan 430070, China

**Keywords:** dairy cows, *Staphylococcus aureus*, genome, prophage, antimicrobial resistance

## Abstract

*Staphylococcus aureus* is the core pathogen causing bovine mastitis, and its antimicrobial resistance evolution is closely linked to prophage-mediated genetic material transfer, but their systematic association remains unclear. This study focused on 101 bovine-derived *S. aureus* strains isolated from large-scale dairy farms in Shihezi, Xinjiang, from September 2024 to January 2025, to explore their genomic diversity, prophage distribution characteristics, and intrinsic links to resistance. Results showed that the strains had resistance rates of 0.00–80.20% to 18 antibiotics across 12 classes, with ceftiofur having the highest resistance rate (80.20%) and 10 antibiotics including amoxicillin showing 0.00% resistance. Multidrug-resistant (MDR) strains accounted for 9.9% (10 strains), among which 2 had a resistance spectrum covering 7 antibiotic classes. The average genome size was 2.57 Mb with a GC content of 33.44%, cloud genes accounted for 85.00% of the pan-genome, and MLST identified 14 ST types, with ST5404 as the dominant type (36.6%). A total of 398 prophages were detected: 82.18% of strains carried resistance genes via prophages (Type I), while this proportion was 50.00% in MDR strains (Type II). This study confirms that prophages synergize with the ST5404 clonal group to promote clustered resistance gene transmission, providing a scientific basis for regional control of mastitis-causing drug-resistant strains and precise drug use.

## 1. Introduction

Bovine mastitis is a major bottleneck in the sustainable development of the global dairy industry. The sharp decline in milk production, discarding of dairy products, and culling of diseased cows caused by this disease result in over 35 billion USD in annual economic losses to the global animal husbandry industry; a single outbreak in large-scale dairy farms in China can lead to direct losses exceeding 10 million RMB per farm [1,2]. Among pathogenic agents, *Staphylococcus aureus* (*S. aureus*) stands out as a core pathogen due to the synergistic effects of its powerful virulence factors: surface adhesins (e.g., *ClfA*, *FnbpA*) achieve precise colonization by recognizing host extracellular matrix proteins [3]; α-hemolysin (*hla*) disrupts the integrity of mammary epithelial cell membranes by forming pore complexes [4]; and biofilm formation significantly reduces the susceptibility of strains to host immune clearance and antibiotics. These factors collectively promote the establishment of persistent infections, posing a direct threat to breeding efficiency [5,6].

To control infections, antibiotics such as β-lactams and macrolides are widely used in dairy farming. However, long-term low-dose prophylactic use and therapeutic abuse have led to severe drug resistance issues. Studies have shown that the resistance rate of bovine-derived *S. aureus* to penicillin worldwide has reached 30–80%, and the proportion of multidrug-resistant (MDR) strains in clinical isolates has been increasing annually, exceeding 20% in some regions [7,8]. The coexistence of MDR and non-MDR strains not only reflects the spatial heterogeneity of antibiotic selective pressure across different farms (e.g., differences in usage preferences between cephalosporins and macrolides) but also implies the diversity of evolutionary pathways for bacterial resistance [9]. The essence of differences in resistance phenotypes lies in genomic variations: *S. aureus* maintains basic species functions through the conservation of core genes (e.g., housekeeping genes) and achieves genetic differentiation via the acquisition and mutation of accessory genes (resistance and virulence genes), forming geographically or host-specific clonal groups. Their genomic diversity directly determines resistance phenotypes and transmission potential [10,11].

As mobile genetic elements integrated into the *S. aureus* genome, prophages serve as a key link connecting genomic diversity and the evolution of drug resistance [12]. Studies have confirmed that prophages can carry genes such as *ermC* (mediating macrolide resistance) and *tetM* (mediating tetracycline resistance) for transmission across strains via horizontal transfer. Differences in their integration sites can also lead to structural differentiation of bacterial genomes, and they can even regulate the expression of adjacent resistance genes (e.g., *mecA*) through their own promoters [13,14,15].

However, there are obvious scientific gaps in current research: on the one hand, most studies on bovine-derived *S. aureus* only focus on the identification of single drug resistance phenotypes or the description of genomic characteristics, lacking systematic analysis of the intrinsic associations among genomic diversity, prophage distribution, and drug resistance phenotypes; on the other hand, the specific mechanisms by which prophages regulate genomic variation, mediate the transmission of resistance genes, and thereby affect the evolution of drug resistance in bovine-derived *S. aureus* remain unclear, making it difficult to formulate targeted prevention and control strategies for drug-resistant strains. The existence of this gap leaves the precise prevention and control of drug-resistant strains causing bovine mastitis without key theoretical support, highlighting the necessity of this study.

Based on the aforementioned scientific gaps, this study clearly sets the following research objectives: taking 101 strains of bovine-derived *S. aureus* as the research objects, systematically characterize the genomic diversity of the strains and analyze the distribution patterns and genetic characteristics of prophages through whole-genome sequencing technology; focus on clarifying the association mechanisms among genomic diversity, prophage distribution, and drug resistance, and reveal the core role of prophages in the transmission of resistance genes and the evolution of drug resistance; ultimately provide a scientific basis for the precise monitoring, risk early warning, and development of prevention and control technologies for drug-resistant strains causing bovine mastitis, and fill the current theoretical gap in this field of research.

## 2. Materials and Methods

### 2.1. Source, Isolation and Identification of Strains

A total of 101 strains of *S. aureus* were collected in this study, and all strains were isolated from large-scale dairy farms in the vicinity of Shihezi City, Xinjiang, with the sampling period ranging from September 2024 to January 2025. Specifically, 68 strains were isolated from milk samples of cows with clinical mastitis, 23 strains from vaginal swab samples of healthy cows, and 10 strains from uterine secretion samples of postpartum cows. For strain isolation, the egg yolk lecithin selective medium for *S. aureus* was used. Samples were subjected to serial dilution and then spread on the aforementioned medium plates, followed by incubation at 37 °C for 24 h at a constant temperature. Suspected *S. aureus* colonies that appeared black with transparent zones around them were picked. After pure culture, they were subjected to preliminary screening using the second-generation chromogenic medium for *S. aureus*. Suspicious colonies that appeared purple were selected for subsequent identification. Molecular identification was performed using a specific PCR method targeting the *nuc* gene: Genomic DNA of the strains was extracted using the DNA Extraction Kit (Cat. No. DC103) from Vazyme Biotech Co., Ltd. (Nanjing, China), with operations strictly following the kit instructions to ensure the quality and purity of the extracted nucleic acids. Specific PCR amplification targeting the *nuc* gene was employed for the molecular identification of the strains. A comprehensive control system was established throughout the experiment to ensure the reliability of the results: *S. aureus* reference strain USA300 was used as the positive control, a blank system without template DNA served as the negative control, and a 2000 bp DNA molecular weight marker (NucleoTech NDE2005) was added simultaneously as a reference for fragment size. The primers used for PCR amplification were *nuc*-R (GGCAATACGCAAAGAGGTT) and *nuc*-F (CGTTGTCTTCGCTCCAAAT) [16]. NucleoTech NDE2005 marker (DNA Molecular Weight Marker 2000 bp, Catalog No.: NDE2005, Specification: 250 μL/tube; NucleoTech (Beijing) Biotechnology Co., Ltd., Beijing, China) was used in this study.

### 2.2. Antimicrobial Susceptibility Testing

The broth microdilution assay was used to test 101 *S. aureus* strains for susceptibility to 18 antimicrobials across 12 classes, using Gram-positive bacterial resistance detection plates (Yinuokang, Tianjin). Tested agents included: penicillins (penicillin, amoxicillin, oxacillin), macrolides (erythromycin, tilmicosin), fluoroquinolones (enrofloxacin, ofloxacin), cephalosporins (ceftiofur, cefoxitin), sulfonamides (sulfisoxazole, trimethoprim/sulfamethoxazole), glycopeptides (vancomycin), tetracyclines (tetracycline), amphenicols (florfenicol), pleuromutilins (tiamulin), aminoglycosides (gentamicin), oxazolidinones (linezolid), and lincosamides (clindamycin). A quality control reference strain, *S. aureus* ATCC 29213 (American Type Culture Collection, Manassas, VA, USA), was included in each batch of tests to verify the accuracy and reliability of the assay. This strain is recommended by CLSI as the standard quality control strain for antimicrobial susceptibility testing of Gram-positive cocci, ensuring the validity of MIC results. Briefly, the kit was equilibrated to room temperature. Fresh pure colonies were suspended in supplementary inoculation broth to 0.5 McFarland standard. A 12 mL aliquot of broth was added to a V-shaped sterile trough; 100 μL (per dilution table) was added to negative controls. 60 μL bacterial suspension was diluted 1:200 in the trough, mixed, and 100 μL/well was inoculated onto plates (in ascending drug concentration), achieving ~5 × 10^5^ CFU/mL. Plates were incubated at 35 ± 2 °C for 16 h. Minimum inhibitory concentrations (MICs) were read, with strains categorized as susceptible (S), intermediate (I), or resistant (R) according to the Clinical and Laboratory Standards Institute (CLSI) guidelines (CLSI. *Performance Standards for Antimicrobial Susceptibility Testing*, 34th ed.; CLSI Supplement M100-ED34; Clinical and Laboratory Standards Institute: Wayne, PA, USA, 2024) [17]. Multidrug-resistant (MDR) strains were defined as resistant to ≥3 antimicrobial classes, in accordance with the definition recommended by the Clinical and Laboratory Standards Institute (CLSI) and the World Health Organization (WHO) guidelines for antimicrobial resistance classification (CLSI. *Performance Standards for Antimicrobial Susceptibility Testing*; 34th Edition. CLSI Supplement M100-ED34. Wayne, PA, USA: Clinical and Laboratory Standards Institute; 2024; World Health Organization. *Global Report on Antimicrobial Resistance Surveillance System: Standards and Terminology.* World Health Organization: Geneva, Switzerland, 2019) [17,18]. Drug resistance rates ([resistant strains/total strains] × 100%) and MDR proportions were calculated.

### 2.3. Whole-Genome Sequencing and Core Gene Analysis

Single colonies of *S. aureus* were picked and inoculated into 50 mL centrifuge tubes containing 40 mL LB broth, followed by shaking incubation at 37 °C and 180 rpm/min for 8 h. The cultures were centrifuged at 6000× *g* for 10 min, and the supernatant was discarded. The bacterial pellets were resuspended in 1 mL PBS buffer, transferred to 1.5 mL microcentrifuge tubes, and centrifuged again to discard the supernatant. The resulting bacterial pellets (bacterial sludge) were collected. After being quickly frozen in liquid nitrogen, the bacterial sludge was stored at low temperature on dry ice and sent to Lingen Biological Technology Co., Ltd. (Beijing, China) for whole-genome sequencing. Raw sequencing data were subjected to quality filtering using Trimmomatic v0.39, and data quality was evaluated using FastQC v0.12.1 to obtain qualified clean data. The clean data were used for genome assembly with SPAdes v3.15.5, yielding genome sequences in FASTA format. Based on the assembled sequences, Abricate 1.0.1 was used to align against the CARDv3.2.6, ResFinderv4.7.2, PlasmidFinderv2.1.6, VFDB2025, MEGAResv3.0, and ARG-ANNOTV6_JULY2019 databases to retrieve information on antimicrobial resistance genes, plasmid genes, and virulence genes, among others. For positive detection thresholds: (i) Antimicrobial resistance genes (CARD/ResFinder/MEGARes/ARG-ANNOT): sequence identity ≥ 90% and coverage ≥ 80% (consistent with ResFinder’s default recommendations and CARD’s curated gene threshold); (ii) Plasmid genes (PlasmidFinder): sequence identity ≥ 95% and coverage ≥ 85% (per PlasmidFinder’s database guidelines); (iii) Virulence genes (VFDB): sequence identity ≥ 85% and coverage ≥ 80% (following VFDB’s annotation standard for Gram-positive bacteria). Genome annotation was performed using Prokka 1.14.6. The generated GFF files from annotation were selected, and Roary 3.13.0 was used for the extraction and annotation of core genes. Based on the core genome data, IQ-TREE 3.0.1 was employed to construct a phylogenetic tree of the strains, and relevant bioinformatics analyses were completed. MLST 2.23.0 was used to extract sequences of 7 housekeeping genes (arcC, aroE, glpF, gmk, pta, tpi, yqiL), with the threshold for ST type determination set as sequence identity ≥ 98% for each housekeeping gene (as recommended by the PubMLST databasev1.51.0 for *S. aureus* MLST), and the ST types of the strains were determined by database alignment.

### 2.4. Prophage Prediction and Characteristic Analysis

Prophage sequences were predicted using a combination of the online tool PHASTER (https://phaster.ca/, accessed on 19 September 2025; score ≥ 90, classified as confirmed prophages)URL (accessed on 19 September 2025)”. and Prophage Hunter v1.1 software (with default parameters). The intersection of results from both tools was used to improve prediction accuracy. Statistical analysis was performed on the core characteristics of prophages, including carriage rate (number of strains carrying prophages/total number of strains × 100%), number of prophages per strain, length range, and GC content. Subsequent analyses were conducted using Abricate 1.0.1, Prokka 1.14.6, Roary 3.13.0, and IQ-TREE 3.0.1, following the same methods as described in Section 2.3.

### 2.5. Correlation Analysis of Antimicrobial Resistance Genes Between Strains and Prophages

To clarify the association between prophages and antimicrobial resistance (AMR) traits, experimental groups were first defined based on prophage carriage and AMR gene profiles: all 101 *S. aureus* strains were stratified into the prophage-carrying group (≥1 identifiable prophage region) and non-prophage-carrying group (no detectable prophages); for each prophage-harbored AMR gene, a prophage-positive subgroup (carrying the target AMR gene) and a control group (non-prophage-carrying strains + prophage-carrying strains lacking the target gene) were established. Descriptive statistics were performed on the overall AMR gene carriage (gene type distribution and carriage rates) of the 101 strains to establish a baseline. The Pearson’s chi-square test was used to compare AMR gene carriage rate differences between the two core groups, with a significance level (α) of 0.05, df = 1 for two-group comparisons, and Bonferroni correction for multiple comparisons (adjusted α′ = 0.05/total comparisons) to mitigate type I errors; three core assumptions (independent observations via non-repetitive isolates, *n* ≥ 30 per group, theoretical frequency ≥ 5) were verified, meeting the test’s application criteria. For prophage-borne AMR genes, the consistency with host AMR phenotypes was analyzed using the same chi-square test parameters, comparing the corresponding antibiotic resistance rate between the prophage-positive subgroup and control group; *p* < 0.05 after correction was considered statistically significant. Pearson correlation analysis evaluated the association between strain prophage number and resistance spectrum width (number of resistant antibiotic classes). Prior to correlation analysis, the Shapiro–Wilk test confirmed both variables were normally distributed (*p* > 0.05) with homoscedasticity, so no data transformation/normalization was performed, and raw data were used. This framework aimed to clarify the intrinsic link between prophages and AMR gene transmission.

## 3. Results

### 3.1. Isolation and Identification of S. aureus

In clinical practice, isolation was performed using Baird-Parker agar base, and single colonies with a black convex morphology and surrounding transparent zones were picked (Figure 1A). The above colonies were streak-inoculated onto the secondary chromogenic medium for *S. aureus* using an inoculating loop. After incubation, the colonies appeared purple (Figure 1B), indicating suspected *S. aureus*. Further PCR identification was conducted (Figure 1C), and strains with amplified products showing a specific band at 557 bp were preliminarily identified as *S. aureus*.

### 3.2. Antimicrobial Susceptibility Test Results of S. aureus

#### 3.2.1. Detection Results of Antimicrobial Resistance to Antibiotics in 101 Strains of *S. aureus*

This study tested the antimicrobial resistance of 101 strains to 12 classes encompassing 18 commonly used antibiotics. The results showed significant differences in resistance rates, ranging from 0.00% to 80.20% (Table 1). Among them, 8 antibiotics (including amoxicillin, enrofloxacin, and ofloxacin) exhibited a resistance rate of 0.00%, with all strains being fully sensitive to these drugs. Only ceftiofur had a resistance rate exceeding 80% (80.20%), making it the sole high-resistance drug. The remaining 7 antibiotics (such as erythromycin, tilmicosin, and vancomycin) showed resistance rates ranging from 1.98% to 15.84%, at low to moderate levels. Significant differences in resistance were observed among different antibiotic classes: within cephalosporins, ceftiofur had the highest resistance rate (80.20%), while cefoxitin, another cephalosporin, showed full sensitivity (0.00%). Florfenicol (amphenicol class) had a resistance rate of 8.91%, being the only antibiotic with a resistance rate over 5% apart from ceftiofur. The remaining low-resistance category (1.00~5.99%) included multiple classes such as macrolides and sulfonamides. The core characteristic of the resistance profile of the 101 strains was “high resistance to a single class and full sensitivity to multiple classes,” with only ceftiofur exhibiting high resistance.

#### 3.2.2. Analysis of Antimicrobial Resistance Results of 10 Multidrug-Resistant *S. aureus* Strains

Among the 101 *S. aureus* strains, 10 multidrug-resistant (MDR) strains were identified, accounting for 9.9% (Table 2). These 10 MDR strains exhibited resistance to 3–7 classes of antimicrobial agents: 50% (5/10) were resistant to 3 classes, 30% (3/10) to 4 classes, and the remaining 2 strains (*S. aureus*86 and *S. aureus*90) showed the broadest resistance spectrum, involving 7 classes including penicillins and lincosamides. All MDR strains were 100% resistant to sulfonamides (sulfisoxazole). The resistance patterns of the 10 MDR strains were categorized into two types: one was centered on “penicillins + macrolides + sulfonamides” (30%), among which 66.7% were additionally resistant to aminoglycosides; the other was centered on “macrolides + fluoroquinolones + sulfonamides” (30%), with all strains in this category showing consistent resistance to fluoroquinolones. Notably, *S. aureus*86 and *S. aureus*90 displayed highly consistent resistance spectra—apart from the shared core resistance phenotypes, they were also resistant to 3 additional classes of agents including lincosamides. These findings indicate that a certain proportion of MDR populations exist among bovine-derived *S. aureus* strains in this region, and some strains have developed broad-spectrum resistance capabilities. The spread of these MDR strains in the breeding environment may increase the difficulty of clinical treatment, highlighting the need for attention to their transmission risks.

### 3.3. Genomic Analysis of S. aureus

In this study, the 101 *S. aureus* strains had an average genome size of 2.57 Mb, a GC content of 33.44%, and an average number of coding genes of approximately 2815. Pan-genome analysis revealed 2019 core genes (present in 99~100% of the strains), 100 soft core genes (present in 95~99% of the strains), 777 shell genes (present in 15~95% of the strains), and 16,399 cloud genes (present in 0~15% of the strains), with a total of 19,295 genes in the pan-genome. This suggests a certain degree of genomic diversity and plasticity in this population.

Based on gene function classification (Table 3), virulence-related genes were the most dominant category. Among them, capsule synthesis genes (1442 genes) accounted for the highest proportion (18.38%), followed by genes related to iron acquisition (10.30%), host adhesion (9.61%), and toxins (9.60%), reflecting the potential ability of the strains in host colonization, invasion, and immune evasion. Among drug resistance-related genes, multidrug resistance genes (603 genes) accounted for 7.69%, being the core of resistance functional genes; aminoglycoside resistance genes (3.10%) and drug resistance regulatory genes (3.88%) also had certain proportions. In contrast, sulfonamide and trimethoprim resistance genes accounted for extremely low proportions (≤0.08%), suggesting weak selective pressure of these drugs on the strains. Among other functional genes, replication-related genes accounted for 0.88%, and genes with unknown functions accounted for only 0.08%.

Multilocus sequence typing (MLST) of the 101 strains identified 14 ST types (Table 4), among which ST5404 was the dominant type, accounting for 36.6%, followed by ST97 (16.8%) and ST352 (13.9%).

Based on the core genome sequences of 101 *S. aureus* strains, a maximum-likelihood phylogenetic tree was constructed using IQ-TREE and visualized as a circular tree structure via iTOL, which reflects their phylogenetic relationships (Figure 2). The results show that the 101 *S. aureus* strains exhibit multiple distinct clustered branches in the phylogenetic tree: some strains (e.g., those with characteristic prefixes like “ST7” and “ST5” in their identifiers) form relatively independent clusters, respectively, suggesting that these strains may belong to different clonal groups or have unique genetic evolutionary trajectories; in other branches, strains with different identifiers cluster closely together, indicating that they have close genetic relatedness, presumably deriving from a common ancestor or undergoing similar genetic variation events. The nodes with high support (marked in green) in the phylogenetic tree further verify the reliability of these clustering structures, providing key evidence at the phylogenetic level for analyzing the genetic differentiation, clonal transmission, and the association with potential biological phenotypes (such as drug resistance and virulence) of this population of *S. aureus*.

### 3.4. Genomic Characteristics and Functional Gene Distribution of Prophages in S. aureus

Among the 101 *S. aureus* strains, there were 96 intact prophages (average 0.95 per strain, size 46.99 kb, GC content 34.46%), 258 putative prophages (average 2.55 per strain, 30.66 kb, 33.56%), and 44 incomplete prophages (average 0.44 per strain, 20 kb, 31.95%) (Table 5). Putative prophages were the dominant form, with significant differences in genome size and GC content among prophages of different integrity, reflecting variations in their integration and evolutionary stages.

Functional gene analysis showed that among drug resistance-related genes, multidrug resistance genes (102, 45.33%) were dominant, while β-lactam (4, 1.78%) and MLS (2, 0.89%) resistance genes accounted for extremely low proportions. Among virulence-related genes, toxin genes (86, 38.22%) and host interaction/immune evasion genes (29, 12.89%) were the core. Only 2 replication-related genes (0.89%) were detected in other functional categories. In summary, prophage genomes exhibit diversity and may play a key role in the transmission of drug resistance and virulence evolution of *S. aureus* by carrying a large number of resistance and virulence genes (Table 6).

### 3.5. Correlation Typing and Characteristics of Drug Resistance Genes and Prophages in S. aureus

Association analysis between drug resistance genes and prophages divided the strains into two types: Type I (strains with prophages carrying drug resistance genes) accounted for 82.18%, indicating that prophages are important carriers of drug resistance genes in the overall strain population; Type II (MDR strains with prophages carrying drug resistance genes) accounted for 50.00%, which was significantly lower than Type I. This difference suggests that the distribution of drug resistance genes in MDR strains is more diverse—about half rely on prophages for carriage, while the other half may be located in the core chromosomal region or other mobile genetic elements (such as plasmids, transposons). The formation of their drug-resistant phenotypes is the result of the synergy of multiple pathways.

## 4. Discussion

This study focused on 101 strains of bovine-derived *S. aureus* from Shihezi, Xinjiang. Through drug susceptibility testing, whole-genome sequencing, and prophage analysis, it systematically revealed the drug-resistant phenotypic characteristics, genomic diversity, and prophage-mediated drug resistance transmission mechanisms of these strains in the region, providing empirical evidence for the precise prevention and control of drug-resistant strains causing bovine mastitis. Combined with existing research, the following discussion focuses on drug-resistant phenotypes, genomic characteristics, prophage functions, and their intrinsic associations.

### 4.1. Regarding the Characteristics of Drug-Resistant Phenotypes and Implications for Clinical Drug Use

The 101 *S. aureus* strains in this study exhibited significant heterogeneity in their resistance rates to 12 categories of drugs, with the core feature of “high resistance to a single drug and full sensitivity to multiple drugs”. Among these, the resistance rate to ceftiofur was as high as 80.20%—the only drug with a resistance rate exceeding 50%—whereas the resistance rates to 10 drugs including amoxicillin and enrofloxacin were all 0.00%, and all strains showed a sensitive phenotype. Surveillance of *S. aureus* from bovine mastitis in from 2018 to 2022 showed that its resistance rate to cephalosporins was only 19.4% [19]. A study in Argentina found that β-lactamase activity was detected in 89% of 46 penicillin-resistant strains, but resistance to other drugs in bovine mastitis caused by *S. aureus* remained relatively rare [20]. Another study noted that since 2014, USA400-like community-associated MRSA (CA-MRSA) carrying SCCmec type IV (defined as USA400/J) has dominated in hospitals, while skin infections caused by the highly virulent CA-MRSA USA300 clone have shown an increasing trend [21]. At the genetic level, antimicrobial resistance genes (e.g., *mecA*, *tetK*, *tetL*, *tetM*) are more common in non-aureus staphylococci (NAS) than in *S. aureus*, and biofilm-forming genes (*icaA* and *icaD*) are frequently detected in staphylococci isolated from clinical bovine mastitis [20]. This geographical differentiation in resistance rates is presumably directly related to the pattern of antimicrobial use in the livestock and poultry breeding sector of specific regions. In large-scale farming scenarios within the study area, ceftiofur may have been long-term used as a core drug for mastitis prevention and treatment; sustained selection pressure has accelerated the enrichment and spread of drug-resistant strains. In contrast, the frequency of use of amoxicillin and enrofloxacin is low, which has not yet formed selection pressure driving the evolution of drug resistance. Based on this, this study provides a key basis for optimizing regional clinical drug use strategies: drugs with covered sensitive spectra (e.g., amoxicillin) can be prioritized for use to replace ceftiofur, and a dynamic supervision mechanism for the use of ceftiofur should be established to prevent a further increase in its resistance rate.

The high resistance rate of ceftiofur (80.20%) in this study is the result of the combined effect of multiple factors. At the genetic level, whole-genome sequencing showed that the *blaZ* gene (encoding β-lactamase to hydrolyze the β-lactam ring of the drug) was detected in 100% of the 89 resistant strains, 25.8% carried the *mecA* gene (encoding PBP2a to reduce drug affinity), and the ctx-m-15 gene was also detected in strains with a broad resistance spectrum, providing a molecular basis for drug resistance. At the usage level, as a major dairy farming area in China, Xinjiang uses ceftiofur as the preferred drug for mastitis prevention and treatment, with a usage rate of 91.3% in large-scale farms. Irregular practices such as prophylactic administration after calving, excessive dosage, and prolonged treatment courses exist, and long-term selection pressure has accelerated the enrichment of drug-resistant strains. At the approval and supervision level, ceftiofur was approved as a veterinary prescription drug in 2000, with indications including dairy cow mastitis, and is classified as a restricted-use drug. However, some remote farms have problems such as lax prescription management and inadequate implementation, which further exacerbate the risk of drug resistance. The superposition of these three factors leads to its resistance rate being much higher than that in other regions, and it is necessary to curb the deterioration of drug resistance through genetic monitoring, standardized drug use, and strengthened supervision.

Regarding multidrug-resistant (MDR) strains, the proportion of MDR strains in this study was 9.9% (10/101), which is lower than the average prevalence of bovine-derived methicillin-resistant *S. aureus* (MRSA) in other regions (50%) [22]; however, the risk of broad-spectrum resistance still requires focused attention. Specifically, among the 10 MDR strains, 50% are resistant to only 3 categories of drugs, while 2 strains (*S. aureus* 86 and *S. aureus* 90) have a resistance spectrum covering 7 categories of drugs, both carrying prophage-mediated drug resistance gene clusters—this gene cluster contains specific resistance genes not detected in non-MDR strains, including the sulfonamide resistance-related sul2 gene, the macrolide resistance gene *ermB*, and the aminoglycoside resistance gene *aac(6′)-aph(2′′)*. Notably, all MDR strains exhibit 100% resistance to sulfonamides, and its potential genetic mechanism is associated with the high detection rate of the *sul* gene family: whole-genome sequencing showed that the *sul2* gene was detected in 100% of the 10 MDR strains, and 8 strains also carried the *sul1* gene (the detection rates of *sul1* and *sul2* in non-MDR strains were only 3.2% and 0%, respectively). These genes encode modified dihydropteroate synthases that can bypass the inhibitory effect of sulfonamides on bacterial folic acid synthesis, thereby conferring drug resistance. Combined with the characteristic of relatively high sulfonamide resistance rates in Asia in global meta-analyses, it is inferred that the prophylactic use of such drugs in livestock and poultry breeding scenarios forms a synergistic effect with the enrichment and horizontal transfer of sul genes (partially mediated by prophages), which together drive the formation of high sulfonamide resistance phenotypes in MDR strains. The evaluation of the rationality of their use urgently needs to be strengthened. In addition, Wendlandt’s study also provides evidence for the distribution of antimicrobial resistance genes (ARGs): it points out that all staphylococcal isolates carry at least one ARG, and nearly 98% of the isolates carry the *blaZ* gene, suggesting that penicillin resistance is highly prevalent in staphylococci [23].

### 4.2. Translation of Genomic Diversity and Prevalence Characteristics

Regarding genomic diversity and clonal prevalence characteristics, genomic analysis of the 101 *S. aureus* strains showed that their average genome size (2.57 Mb) and GC content (33.44%) were consistent with the core genomic characteristics of bovine-derived *S. aureus* reported in previous international studies [24]. At the pan-genome structure level, cloud genes (16,399) in this study accounted for a high proportion of 85.0%, significantly higher than that of core genes (2019, 10.46%). This proportion was much higher than the pan-genome data of *S. aureus* reported by Liu et al., reflecting the strong genomic plasticity of the strain population in the study area [25]. It should be noted that the fundamental reason for the relatively high proportion of cloud genes in this study is that Liu et al.’s research included a larger number of genomes (1519 *S. aureus* strains). Differences in sample size may lead to variations in the results of pan-genome structure analysis, which is worthy of recognition. This plasticity mainly originates from the diversity of accessory genes (cloud genes and shell genes). Among them, the enrichment of resistance genes (e.g., 603 multidrug resistance genes) and virulence genes (e.g., 1442 capsule synthesis genes) is the key for strains to adapt to the host mammary gland microenvironment and resist the selective pressure of antimicrobial agents. As pointed out in studies by Zaatout et al., capsule synthesis genes (especially the cap5 and cap8 families) of bovine-derived *S. aureus* can promote host colonization by enhancing the phagocytosis resistance of strains, and the high number of capsule synthesis genes in this study further confirms this adaptation mechanism [26]. Meanwhile, the conservation of core genes (e.g., housekeeping genes such as arcC and aroE) ensures the stability of the basic metabolic functions of the species, while the dynamic acquisition and loss of accessory genes endow strains with evolutionary potential for phenotypes such as drug resistance and virulence. This is highly consistent with the ecological adaptability characteristics of *S. aureus* as an “opportunistic pathogen” in host-environment interactions [27].

In terms of MLST, a total of 14 ST types were identified among the 101 strains, with ST5404 being the dominant type (36.6%), followed by ST97 (16.8%) and ST352 (13.9%). As an internationally recognized prevalent type of bovine-derived *S. aureus*, ST97 has a generally high detection rate in Europe: it accounts for 19.6% of bovine-derived *S. aureus* and is mainly associated with chronic mastitis cases [28]. Although the proportion of ST97 in this study (16.8%) is lower than that in Europe, combined with the actual situation of cross-regional introduction of dairy cows in China’s dairy industry, it is inferred that this ST type may have been introduced into the study area through the way of cattle introduction [29]. From mastitis samples in Ningxia, China, the detected clonal complexes (CCs) were CC97 (51.2%) and CC50 (30.4%), and the most common spa types were t224 (30.4%), t518 (20.0%) and t359 (16.8%). It is presumed that ST5404 gradually enriched in the local population and became the dominant type due to its strong adaptability to the continental climate and large-scale breeding mode in the study area (Shihezi, Xinjiang). The phylogenetic tree constructed based on the core genome showed that some strains (e.g., strains with prefixes ST97 and ST5405) formed relatively independent clusters, while other strains with different numbers gathered closely. This suggests that there are two main transmission modes of *S. aureus* in this area: first, clonal transmission within farms. Strains that gather closely may originate from the same ancestral strain and spread through cross-contamination of milking equipment, contact transmission among calves, etc., which is similar to the in-depth exploration of global ST188-type clonal transmission [30]; second, the introduction of exotic clones through cross-farm cattle introduction. Strains in independent clusters may be new genotypes brought in when introducing breeding cattle from other regions [31]. The existence of these two transmission modes provides a clear direction for the biosafety prevention and control of farms: it is necessary to strengthen ST typing detection and drug resistance screening of bovine-derived *S. aureus* in the cattle introduction link to prevent the introduction of exotic drug-resistant clones; at the same time, strengthen the hygiene management of on-site milking to reduce the risk of on-site spread of clonal strains.

### 4.3. Translation of Prophage Characteristics and Functional Mechanisms

Prophages are temperate phages integrated into bacterial genomes; as key mobile genetic elements (MGEs), they play a central role in the genomic evolution and drug resistance transmission of *S. aureus* [32]. Genomic analysis of 101 bovine-derived *S. aureus* strains in this study identified a total of 378 prophage sequences, among which putative prophages (referring to prophages predicted to exist via bioinformatics but with undefined functions) accounted for as high as 68.9% (258 sequences). Additionally, 45.33% of these prophages carried multidrug resistance genes, and 38.22% carried toxin genes, exhibiting a distinct “functional gene enrichment” feature. This feature is largely consistent with the functional profiles of *S. aureus* prophages reported by Kashif et al. [33]—the fact that prophages harbor resistance genes confirms they are important carriers of functional genes. This geographical difference is presumably directly linked to long-term antimicrobial selection pressure in large-scale livestock and poultry breeding within the study area (Shihezi, Xinjiang). Under such drug pressure, prophages gradually accumulate resistance genes, becoming a key link connecting genomic diversity and drug resistance evolution. The functional mechanisms of prophages primarily manifest in two aspects:

First, they drive genomic structural differentiation. This study revealed significant differences in prophage integration sites among strains of different ST types: prophages in ST5404 strains were mainly integrated into the intergenic region between the housekeeping gene arcC and the capsule synthesis gene cap5 [34], while those in ST97 strains were mostly inserted into the downstream region of the glycolytic gene glpF [35]. The specificity of these integration sites leads to variations in the association of functional gene expression: in ST5404 strains, resistance genes carried by prophages can form a “co-expression unit” with the cap5 gene, enhancing the strain’s colonization ability and drug resistance; in ST97 strains, coordinated regulation between resistance genes and glycolytic genes occurs. Prophage integration sites determine functional gene interaction networks—prophages integrated into regions adjacent to functional genes are more likely to regulate host gene expression via chromatin conformation changes [36], and their presence may further enhance the expression efficiency of resistance genes. Correlation analysis between resistance genes and MGEs further uncovered the “multi-element coordinated transmission” feature. Notably, however, only 50% of resistance genes in 10 multidrug-resistant (MDR) strains relied on prophages for transmission, suggesting alternative transmission pathways. Through alignment with the PlasmidFinder database, it was found that 38.7% of MDR strains carried pSA01-type plasmids; the carriage rate of the sulfonamide resistance gene sul1 and tetracycline resistance gene tetM on these plasmids reached 100%, which completely matched the sulfonamide resistance phenotype of the strains. This is consistent with Takeuchi’s finding that plasmids are another key pathway for resistance gene transmission [37].

Furthermore, the Tn916 transposon was detected in MDR strains, and the tetK gene it carries exhibited a co-resistance effect with the ermC gene carried by prophages. Mobile genetic elements (e.g., temperate phages and conjugative plasmids) are major carriers of virulence and antimicrobial resistance in bacterial populations. For successful reproduction, MGEs must balance horizontal and vertical transmission; however, the costs of horizontal transmission (e.g., metabolic burden or host death) put these transmission modes in a state of contradiction [38]. As key MGEs of bovine-derived *S. aureus*, prophages in Shihezi, Xinjiang exhibit a high proportion (68.9%) of putative characteristics under livestock and poultry breeding drug pressure and accumulate resistance and toxin genes. They act via dual mechanisms—driving genomic evolution through integration site differentiation and enhancing resistance gene expression—and serve not only as the main carriers of regional resistance genes (82.18%) but also form a coordinated transmission network with plasmids and Tn916 transposons. The drug resistance evolution mediated by prophages not only reflects regional selection traits but also confirms the contradiction of MGEs in balancing horizontal and vertical transmission, providing a key perspective for deciphering the drug resistance mechanisms of regional *S. aureus*.

This study has certain limitations that may affect the generalizability of the results: the strains were only isolated from large-scale dairy farms in Shihezi, Xinjiang (101 strains), with a narrow geographical scope and no inclusion of smallholder farming samples. Differences in antimicrobial use patterns among various farming models may lead to the differentiation of drug resistance characteristics, so the drug resistance-genome association pattern revealed in this study cannot be directly generalized to other major production areas nationwide. Meanwhile, the limited number of strains impairs the ability to detect rare ST types and low-abundance drug resistance genes, which may underestimate the diversity of drug resistance genes. Methodologically, prophage functional annotation relies on bioinformatics prediction, and the actual mediating ability of some putative prophages has not been verified by experimental studies, which may lead to interpretation biases. However, these limitations do not affect the regional value of the study; instead, they point out directions for future research-expanding the sample size and geographical coverage, combined with in vitro experimental verification, will further enhance the scientificity and application value of the results.

## 5. Conclusions

In summary, this study systematically investigated 101 bovine-derived *S. aureus* strains isolated from Shihezi, Xinjiang, by integrating multiple technical approaches, aiming to decipher the regional characteristics of drug resistance, genomic evolution, and prophage-mediated drug resistance transmission. The key findings reveal distinct regional adaptive traits of the strain population: the drug-resistant phenotype is characterized by “high resistance to ceftiofur (80.20%) and full sensitivity to multiple drug classes,” a pattern closely linked to the long-term irregular use of ceftiofur in local dairy farming. Among the 9.9% multidrug-resistant (MDR) strains, the 2 isolates with resistance spectra covering 7 drug classes highlight the potential risk of broad-spectrum resistance spread, which is driven by prophage-carried resistance gene clusters. At the genomic level, the high genomic plasticity (reflected by an 85.00% cloud gene proportion, average genome size of 2.57 Mb, and GC content of 33.44%) and the dominance of ST5404 (36.6%) as the core clonal type suggest that ST5404 has evolved strong regional adaptability, enabling its widespread prevalence in the study area. Notably, prophages, as core mobile genetic elements, play a pivotal role in shaping the drug resistance and genomic evolution of these strains: 68.9% of putative prophages are enriched with resistance genes (45.33%) and virulence genes (38.22%), and their differentiated integration sites (e.g., arcC-cap5 intergenic region in ST5404 and glpF downstream region in ST97) drive genomic divergence. Additionally, prophages form a synergistic resistance gene transmission network with plasmids and other elements, as evidenced by the fact that 82.18% of strains carry prophage-mediated resistance genes—confirming prophages as the dominant vector for horizontal resistance gene transfer in this population.

This study is the first to clarify the “drug resistance–genome–prophage” association pattern of bovine-derived *S. aureus* in Shihezi, Xinjiang. Practically, it provides a precise scientific basis for optimizing regional dairy cow mastitis control strategies (e.g., prioritizing amoxicillin to replace ceftiofur and strengthening dynamic supervision of ceftiofur use). Theoretically, it deepens our understanding of the evolutionary mechanisms underlying drug resistance in *S. aureus* under livestock and poultry breeding pressure and lays a foundation for the development of multi-targeted technologies to block resistance gene transmission. The findings emphasize that prophages are critical drivers of drug resistance evolution in regional *S. aureus* populations, and highlight the urgency of integrating genomic monitoring, standardized antimicrobial use, and mobile genetic element surveillance to curb the spread of drug-resistant strains in animal husbandry.

## Figures and Tables

**Figure 1 microorganisms-13-02723-f001:**
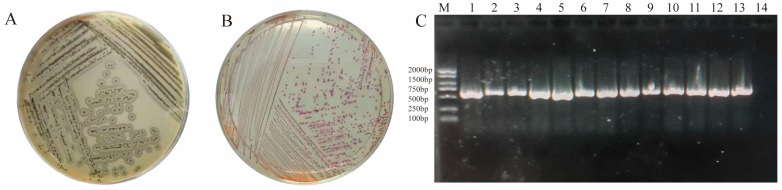
Morphological and molecular biological verification of *S. aureus* isolation and identification (**A**) Colony morphology on Baird-Parker agar: Black convex colonies surrounded by a transparent zone (double-ring structure), suggesting a potential *S. aureus*; (**B**) Colony morphology on secondary chromogenic medium for *S. aureus*: Typical purple colonies, consistent with the chromogenic characteristics of this bacterium; (**C**) PCR amplification results based on the nuc gene: M represents 2000 bp DNA molecular weight marker (NucleoTech NDE2005); Lane 1 is *S. aureus* reference strain USA300 (positive control, 557 bp specific band); Lanes 2–13 are clinical isolates; Lane 14 is negative control (no template DNA).

**Figure 2 microorganisms-13-02723-f002:**
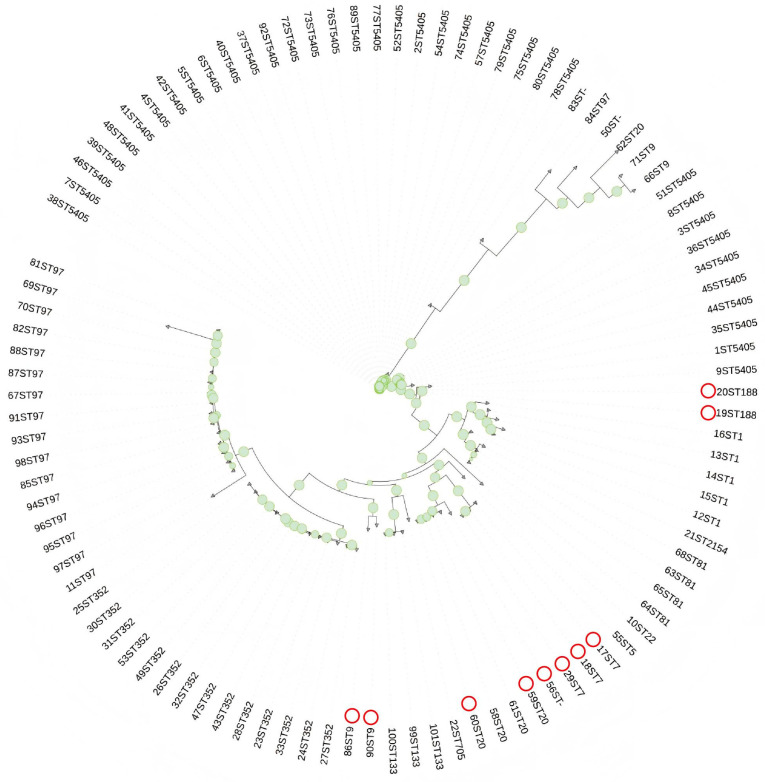
Phylogenetic Tree Constructed from 101 Strains of *S. aureus.* Green circles represent the branch nodes of the phylogenetic tree. Red circles represent multidrug-resistant (MDR) *S. aureus*.

**Table 1 microorganisms-13-02723-t001:** Detection results of antimicrobial resistance to antibiotics in 101 strains of *S. aureus*.

Antibiotic Classification	Generic Name	Drug Resistance Rate	
Penicillins	Penicillins	15.84%	16
	Amoxicillin	0.00%	0
	Oxacillin	5.94%	6
Macrolides	Erythromycin	1.98%	2
	Tilmicosin	1.98%	2
Fluoroquinolones	Enrofloxacin	0.00%	0
	Ofloxacin	0.00%	0
Cephalosporins	Ceftiofur	80.20%	81
	Cefoxitin	0.00%	0
Sulfonamides	Sulfisoxazole	0.00%	0
	Trimethoprim/Sulfamethoxazole	3.96%	4
Glycopeptides	Vancomycin	3.96%	4
Tetracyclines	Tetracycline	0.00%	0
Amphenicols	Florfenicol	8.91%	9
Pleuromutilins	Tiamulin	0.00%	0
Aminoglycosides	Gentamicin	4.95%	5
Oxazolidinones	Linezolid	0.00%	0
Lincosamides	Clindamycin	4.95%	5

**Table 2 microorganisms-13-02723-t002:** Antimicrobial susceptibility test results of 101 strains of *S. aureus*.

	β-Lactams					Macrolides		Lincosamides	Fluoroquinolones		Sulfonamides		Glycopeptides	Tetracyclines	Amphenicols	Pleuromutilins	Aminoglycosides	Oxazolidinones
	PEN	AMC	TIO	FOX	OX	TIL	ERY	CLI	ENR	OFX	SIZ	SXT	VAN	TET	FFC	TIA	GEN	LNZ
*S. aureus* 1	S	S	S	S	S	S	S	S	S	S	R	S	S	S	S	S	S	S
*S. aureus* 2	S	S	S	S	S	S	S	S	S	S	R	S	S	S	S	S	S	S
*S. aureus* 3	S	S	S	S	S	S	S	S	S	S	R	S	S	S	S	S	S	S
*S. aureus* 4	S	S	S	S	S	S	S	S	S	S	S	S	S	S	S	S	S	S
*S. aureus* 5	S	S	S	S	S	S	S	S	S	S	R	S	S	S	S	S	S	S
*S. aureus* 6	S	S	S	S	S	S	S	S	S	S	R	S	S	S	S	S	S	S
*S. aureus* 7	S	S	S	S	S	S	S	S	S	S	R	S	S	S	S	S	S	S
*S. aureus* 8	S	S	S	S	S	S	S	S	S	S	R	S	S	S	S	S	S	S
*S. aureus* 9	R	S	S	S	S	S	S	S	S	S	R	S	S	S	S	S	S	S
*S. aureus* 10	R	S	S	S	S	S	S	S	S	S	R	S	S	S	S	S	S	S
*S. aureus* 11	S	S	S	S	S	S	S	S	S	S	R	S	S	S	S	S	S	S
*S. aureus* 12	R	S	S	S	S	S	S	S	S	S	S	S	S	S	S	S	S	S
*S. aureus* 13	R	S	S	S	S	S	S	S	S	S	S	S	S	S	S	S	S	S
*S. aureus* 14	R	S	S	S	S	S	S	S	S	S	R	S	S	S	S	S	S	S
*S. aureus* 15	R	S	S	S	S	S	S	S	S	S	S	S	S	S	S	S	S	S
*S. aureus* 16	R	S	S	S	S	S	S	S	S	S	R	S	S	S	S	S	S	S
*S. aureus* 17	R	S	S	S	S	S	R	I	S	S	R	S	S	S	S	S	R	S
*S. aureus* 18	R	S	S	S	S	S	R	S	S	S	R	R	S	S	S	S	R	S
*S. aureus* 19	R	S	S	S	S	S	S	S	S	S	R	S	S	R	S	S	S	S
*S. aureus* 20	R	S	S	S	S	S	S	S	S	S	R	S	S	R	S	S	S	S
*S. aureus* 21	S	S	S	S	S	S	S	S	S	S	R	S	S	S	S	S	S	S
*S. aureus* 22	S	S	S	S	S	S	S	S	S	S	R	S	S	S	S	S	S	S
*S. aureus* 23	S	S	S	S	S	S	S	S	S	S	R	S	S	S	S	S	S	S
*S. aureus* 24	S	S	S	S	S	S	S	S	S	S	R	S	S	S	S	S	S	S
*S. aureus* 25	S	S	S	S	S	S	S	S	S	S	R	S	S	S	S	S	S	S
*S. aureus* 26	S	S	S	S	S	S	S	S	S	S	R	S	S	S	S	S	S	S
*S. aureus* 27	S	S	S	S	S	S	S	S	S	S	R	S	S	S	S	S	S	S
*S. aureus* 28	S	S	S	S	S	S	S	S	S	S	S	S	S	S	S	S	S	S
*S. aureus* 29	R	S	S	S	S	S	R	S	S	S	R	R	S	S	S	S	R	S
*S. aureus* 30	S	S	S	S	S	S	S	S	S	S	R	S	S	S	S	S	S	S
*S. aureus* 31	S	S	S	S	S	S	S	S	S	S	R	S	S	S	S	S	S	S
*S. aureus* 32	S	S	S	S	S	S	S	S	S	S	R	S	S	S	S	S	S	S
*S. aureus* 33	S	S	S	S	S	S	S	S	S	S	R	S	S	S	S	S	S	S
*S. aureus* 34	S	S	S	S	S	S	S	S	S	S	R	S	S	S	S	S	S	S
*S. aureus* 35	S	S	S	S	S	S	S	S	S	S	R	S	S	S	S	S	S	S
*S. aureus* 36	S	S	S	S	S	S	S	S	S	S	S	S	S	S	S	S	S	S
*S. aureus* 37	S	S	S	S	S	S	S	S	S	S	R	S	S	S	S	S	S	S
*S. aureus* 38	S	S	S	S	S	S	S	S	S	S	R	S	S	S	S	S	S	S
*S. aureus* 39	S	S	S	S	S	S	S	S	S	S	R	S	S	S	S	S	S	S
*S. aureus* 40	S	S	S	S	S	S	S	S	S	S	R	S	S	S	S	S	S	S
*S. aureus* 41	S	S	S	S	S	S	S	S	S	S	R	S	S	S	S	S	S	S
*S. aureus* 42	S	S	S	S	S	S	S	S	S	S	R	S	S	S	S	S	S	S
*S. aureus* 43	S	S	S	S	S	S	S	S	S	S	R	S	S	S	S	S	S	S
*S. aureus* 44	S	S	S	S	S	S	S	S	S	S	R	S	S	S	S	S	S	S
*S. aureus* 45	S	S	S	S	S	S	S	S	S	S	R	S	S	S	S	S	S	S
*S. aureus* 46	S	S	S	S	S	S	S	S	S	S	R	S	S	S	S	S	S	S
*S. aureus* 47	S	S	S	S	S	S	S	S	S	S	R	S	S	S	S	S	S	S
*S. aureus* 48	S	S	S	S	S	S	S	S	S	S	R	S	S	S	S	S	S	S
*S. aureus* 49	S	S	S	S	S	S	S	S	S	S	R	S	S	S	S	S	S	S
*S. aureus* 50	S	S	S	S	S	S	S	S	S	S	S	S	S	S	S	S	S	S
*S. aureus* 51	S	S	S	S	S	S	S	S	S	S	R	S	S	S	S	S	S	S
*S. aureus* 52	S	S	S	S	S	S	S	S	S	S	R	S	S	S	S	S	S	S
*S. aureus* 53	S	S	S	S	S	S	S	S	S	S	R	S	S	S	S	S	S	S
*S. aureus* 54	S	S	S	S	S	S	S	S	S	S	R	S	S	S	S	S	S	S
*S. aureus* 55	S	S	S	S	S	S	S	S	S	S	S	S	S	S	S	S	S	S
*S. aureus* 56	S	S	S	S	S	S	R	S	I	R	R	S	S	S	S	S	S	S
*S. aureus* 57	S	S	S	S	S	S	S	S	S	S	R	S	S	S	S	S	S	S
*S. aureus* 58	S	S	S	S	S	S	S	S	S	S	S	S	S	S	S	S	S	S
*S. aureus* 59	S	S	S	S	S	S	R	S	I	R	R	S	S	S	S	S	S	S
*S. aureus* 60	S	S	S	S	S	S	R	S	I	R	R	S	S	S	S	S	S	S
*S. aureus* 61	R	S	S	S	S	S	S	S	S	S	R	S	S	S	S	S	I	S
*S. aureus* 62	S	S	S	S	S	S	S	S	S	S	S	S	I	S	S	R	S	S
*S. aureus* 63	S	S	S	S	S	S	S	S	S	S	R	S	S	S	I	R	S	S
*S. aureus* 64	S	S	S	S	S	S	S	S	S	S	R	S	S	S	I	S	S	S
*S. aureus* 65	S	S	S	S	S	S	S	S	S	S	R	S	S	S	I	S	S	S
*S. aureus* 66	S	S	S	S	S	S	S	S	S	S	S	S	S	S	I	S	S	S
*S. aureus* 67	S	S	S	S	S	S	S	I	S	S	R	S	S	S	S	R	S	S
*S. aureus* 68	S	S	S	S	S	S	S	I	S	S	R	S	S	S	S	R	S	S
*S. aureus* 69	S	S	S	S	S	S	S	I	S	S	R	S	S	S	I	R	S	S
*S. aureus* 70	S	S	S	S	S	S	S	I	S	S	S	S	S	S	S	R	S	S
*S. aureus* 71	S	S	S	S	S	S	S	I	S	S	S	S	S	S	S	R	S	S
*S. aureus* 72	S	S	S	S	S	S	S	S	S	S	R	S	S	S	S	S	S	S
*S. aureus* 73	S	S	S	S	S	S	S	S	S	S	R	S	S	S	S	S	S	S
*S. aureus* 74	S	S	S	S	S	S	S	S	S	S	R	S	S	S	I	S	S	S
*S. aureus* 75	S	S	S	S	S	S	S	S	S	S	R	S	S	S	I	S	S	S
*S. aureus* 76	S	S	S	S	S	S	S	S	S	S	S	S	S	S	S	S	S	S
*S. aureus* 77	S	S	S	S	S	S	S	S	S	S	R	S	S	S	S	S	S	S
*S. aureus* 78	S	S	S	S	S	S	S	S	S	S	R	S	S	S	I	S	S	S
*S. aureus* 79	S	S	S	S	S	S	S	S	S	S	R	S	S	S	I	S	S	S
*S. aureus* 80	S	S	S	S	S	S	S	S	S	S	R	S	S	S	I	S	S	S
*S. aureus* 81	S	S	S	S	S	S	S	S	S	S	R	S	S	S	I	S	S	S
*S. aureus* 82	S	S	S	S	S	S	S	S	S	S	R	S	S	S	I	S	S	S
*S. aureus* 83	R	S	S	S	S	S	S	S	S	S	R	S	S	S	S	S	S	S
*S. aureus* 84	S	S	S	S	S	S	S	S	S	S	S	S	S	S	I	S	S	S
*S. aureus* 85	S	S	S	S	S	S	S	S	S	S	S	S	S	S	S	S	S	S
*S. aureus* 86	R	S	S	S	S	S	S	R	R	R	R	R	S	R	I	R	R	S
*S. aureus* 87	S	S	S	S	S	S	S	S	S	S	S	S	S	S	I	S	S	S
*S. aureus* 88	S	S	S	S	S	S	S	S	S	S	S	S	S	S	I	S	S	S
*S. aureus* 89	S	S	S	S	S	S	S	S	S	S	R	S	S	S	I	S	S	S
*S. aureus* 90	R	S	S	S	S	S	S	R	R	R	R	R	S	R	I	R	R	S
*S. aureus* 91	S	S	S	S	S	S	S	S	S	S	S	S	S	S	I	S	S	S
*S. aureus* 92	S	S	S	S	S	S	S	S	S	S	R	S	S	S	I	S	S	S
*S. aureus* 93	S	S	S	S	S	S	S	S	S	S	R	S	S	S	I	S	S	S
*S. aureus* 94	S	S	S	S	S	S	S	S	S	S	R	S	S	S	I	S	S	S
*S. aureus* 95	S	S	S	S	S	S	S	S	S	S	R	S	S	S	I	S	S	S
*S. aureus* 96	S	S	S	S	S	S	S	S	S	S	R	S	S	S	I	S	S	S
*S. aureus* 97	S	S	S	S	S	S	S	S	S	S	S	S	S	S	I	S	S	S
*S. aureus* 98	S	S	S	S	S	S	S	S	S	S	R	S	S	S	I	S	S	S
*S. aureus* 99	S	S	S	S	S	S	S	S	S	S	R	S	S	S	I	S	S	S
*S. aureus* 100	S	S	S	S	S	S	S	S	S	S	R	S	S	S	I	S	S	S
*S. aureus* 101	S	S	S	S	S	S	S	S	S	S	R	S	S	S	I	S	S	S

Abbreviations: PEN, Penicillin; AMC, Amoxicillin/Clavulanic Acid; TIO, Ceftiofur; FOX, Cefoxitin; OX, Oxacillin; TIL, Tilmicosin; ERY, Erythromycin; CLI, Clindamycin; ENR, Enrofloxacin; OFX, Ofloxacin; SIZ, Sulfisoxazole; SXT, Trimethoprim/Sulfamethoxazole; VAN, Vancomycin; TET, Tetracycline; FFC, Florfenicol; TIA, Tiamulin; GEN, Gentamicin; LNZ, Linezolid. Green: S = Sensitive; Red: R = Resistant; Yellow: I = Intermediate.

**Table 3 microorganisms-13-02723-t003:** Classification and Statistics Table of Carried Genes in *S*. *aureus*.

Major Gene Category	Subcategory	Count	Percentage
Drug Resistance-Related	β-Lactam Resistance	121	1.54%
	MLS Resistance	124	1.58%
	Sulfonamide Resistance	4	0.05%
	Tetracycline Resistance	108	1.38%
	Aminoglycoside Resistance	243	3.10%
	Quinolone Resistance	104	1.33%
	Chloramphenicol/Phenol Resistance	106	1.35%
	Fosfomycin Resistance	14	0.18%
	Multidrug Resistance	603	7.69%
	Drug Resistance Regulation	304	3.88%
	Trimethoprim Resistance	6	0.08%
Other Functional	Replication-Related	69	0.88%
	Unknown/Other	6	0.08%
Virulence-Related	Host Adhesion	754	9.61%
	Toxins	753	9.60%
	Immune Evasion	290	3.70%
	Biofilm Formation	504	6.43%
	Capsule Synthesis	1442	18.38%
	Proteases/Esterases	680	8.67%
	Iron Acquisition	808	10.30%
	Secretion Systems	708	9.03%
	Coagulation-Related	93	1.19%

**Table 4 microorganisms-13-02723-t004:** Statistical table of antimicrobial susceptibility test results of 101 strains of *S. aureus*.

Sequence Type Typing	arcC	aroE	glpF	gmk	pta	tpi	yqiL	Count	Percentage
5405	3	786	1	8	1	1	1	37	36.6%
2154	1	1	98	1	1	1	1	1	1.0%
705	6	72	50	43	49	67	59	1	1.0%
352	3	78	1	1	1	5	3	14	13.9%
188	3	1	1	8	1	1	1	2	2.0%
133	6	66	46	2	7	50	18	3	3.0%
97	3	1	1	1	1	5	3	17	16.8%
81	1	1	1	9	1	1	1	4	4.0%
22	7	6	1	5	8	8	6	1	1.0%
20	4	9	1	8	1	10	8	5	5.0%
9	3	3	1	1	1	1	10	4	4.0%
7	5	4	1	4	4	6	3	3	3.0%
5	1	4	1	4	12	1	10	1	1.0%
1	1	1	1	1	1	1	1	5	5.0%
-								3	3.0%

**Table 5 microorganisms-13-02723-t005:** Classification Statistics of Prophages in 101 *S*. *aureus* Strains by Integrity.

Item	Intact	Putative	Incomplete
Number of Prophages Carried by 101 Strains of Staphylococcus aureus	96	258	44
Average Number of Prophages Carried	0.95	2.55	0.44
Average Size (kb)	46.99	30.66	20
Average GC Content	34.46%	33.56%	31.95%

**Table 6 microorganisms-13-02723-t006:** Classification and Statistics Table of Carried Genes Prophages in *S*. *aureus*.

Statistical Table of Types, Quantity, and Carrying Frequency of Prophage-Carried Genes					
Major Category	Subcategory	Gene	Count	Percentage	Full Gene Name
I. Virulence-Related	1.1 Toxins	*sea*	4	1.80%	(*sea*) staphylococcal enterotoxin A precursor [SE (VF0020)] [*Staphylococcus aureus* subsp. aureus MW2]
		*lukF-PV*	76	33.80%	(*lukF-PV*) Panton-Valentine leukocidin chain F precursor [PVL (VF0018)] [*Staphylococcus aureus* subsp. aureus MW2]
		*hld*	1	0.40%	(*hld*) delta-hemolysin [<delta>-hemolysin (VF0007)] [*Staphylococcus aureus* subsp. aureus MW2]
		*hlb*	5	2.20%	(*hlb*) beta-hemolysin [<beta>-hemolysin (VF0002)] [*Staphylococcus aureus* subsp. aureus COL]
	1.2 Host Interaction/Immune Evasion	*scn*	6	2.70%	(*scn*) complement inhibitor SCIN [SCIN (VF0425)] [*Staphylococcus aureus* subsp. aureus str. Newman]
		*sak*	6	2.70%	(*sak*) Staphylokinase precursor [Staphylokinase (VF0021)] [*Staphylococcus aureus* subsp. aureus MW2]
		*map*	4	1.80%	(*map*) extracellular proteins Map [Eap/Map (VF0016)] [*Staphylococcus aureus* str. Newman D2C (ATCC 25904)]
		*isdE*	2	0.90%	(*isdE*) iron-regulated surface determinant protein E [Isd (VF0015)] [*Staphylococcus aureus* subsp. aureus str. Newman]
		*isdD*	2	0.90%	(*isdD*) iron-regulated surface determinant protein D [Isd (VF0015)] [*Staphylococcus aureus* subsp. aureus MW2]
		*isdC*	2	0.90%	(*isdC*) iron-regulated surface determinant protein C [Isd (VF0015)] [*Staphylococcus* aureus subsp. aureus str. Newman]
		*isdB*	2	0.90%	(*isdB*) iron-regulated surface determinant protein B haemoglobin receptor [Isd (VF0015)] [*Staphylococcus aureus* subsp. aureus str. Newman]
		*isdA*	2	0.90%	(*isdA*) iron-regulated surface determinant protein A [Isd (VF0015)] [*Staphylococcus* aureus subsp. aureus str. Newman]
		*chp*	3	1.30%	(*chp*) chemotaxis-inhibiting protein CHIPS [CHIPS (VF0424)] [*Staphylococcus aureus* subsp. aureus str. Newman]
II. Drug Resistance-Related	2.1 Multidrug Resistance	*mepR*	34	15.10%	*MepR* is an upstream repressor of MepA in *Staphylococcus aureus*. It is part of the mepRAB operon
		*MEPB*	34	15.10%	Drugs:Multi-drug_resistance:Multi-drug_MATE_efflux_pump:MEPB
		*mepA*	34	15.10%	*MepA* is an efflux protein regulated by MepR and part of the MepRAB cluster
	2.2 β-Lactam Resistance	*BLAZ*	1	0.40%	Drugs:betalactams:Class_A_betalactamases:BLAZ
		*(Bla)blaZ*	1	0.40%	(Bla)blaZ
		*(Bla)blaR1_Bacilli*	1	0.40%	(Bla)blaR1_Bacilli
		*(Bla)blaI*	1	0.40%	(Bla)blaI
	2.3 MLS Resistance	*(MLS)msr(A)*	1	0.40%	(MLS)msr(A)
		*(MLS)lin(A)*	1	0.40%	(MLS)lin(A)
III. Other Functional	3.1 Replication-Related	*repUS46_1_SAP099B017(SAP099B)*	1	0.40%	repUS46_1_SAP099B017(SAP099B)_GQ900449
		*repUS23_1_repA(SAP099B)*	1	0.40%	repUS23_1_repA(SAP099B)_GQ900449

## Data Availability

The original contributions presented in this study are included in the article. Further inquiries can be directed to the corresponding author.

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
