# Peer review of "Study on Genomic Diversity, Prophage Distribution of Bovine-Derived Staphylococcus aureus and Their Association with Antimicrobial Resistance"

_microorganisms, 2025, doi:10.3390/microorganisms13122723_

Round 1

Reviewer 1 Report

Comments and Suggestions for Authors

This manuscript provides a clear and relevant investigation into antibiotic resistance in bovine-derived S. aureus, combining antimicrobial susceptibility testing, whole-genome sequencing, and prophage analysis to offer an integrated understanding of resistance mechanisms. Furthermore, the findings have practical implications for guiding targeted prevention and control of mastitis in dairy cattle, while laying a foundation for future research on antibiotic resistance evolution.  However, I have some recommendations that should be addressed before the manuscript can be considered for acceptance

Abstract:

Please revise the abstract by substantially reducing its length. The text should focus only on the most important findings. The central contribution of the study should be highlighted more clearly, particularly the relationship between prophages and the evolution of antimicrobial resistance.

Introduction

The introduction should clearly highlight the existing scientific gap, demonstrating what remains unknown and underscoring the necessity of the study. Presenting this gap early helps establish the context and emphasize the significance of the research. The objectives of the study are not clearly defined. I recommend that the authors explicitly state the research goals, clarifying what the study intends to achieve and how it addresses the identified scientific gap.

Material and Methods

Lines 120-123: The description of sample collection could be improved for clarity.  

Line 129: “suspicious colonies” can be replaced with more scientific expressions, for example, “presumptive S. aureus colonies.

Line 129: “second-generation chromogenic medium”- Which one?

Line 151: Please include references for the guidelines followed in the study, as well as for the criteria used to classify strains as multidrug-resistant (MDR).

Please specify the ATCC reference strains used as controls for the MIC assays and for the positive controls in the PCR experiments.

Lines 175-182: The use of Abricate, Prokka, Roary, and IQ-TREE is vague. It is not clear whether these analyses were applied to the prophage regions specifically, or to the entire genomes.

Lines 185-196:

Which statistical test was used in addition to the chi-square test?

How were the groups defined?

Explain the parameters of the chi-square test, adjustments for multiple testing, and verification of its assumptions.

For the Pearson correlation, indicate whether the data were transformed or normalized.

Which analyses were performed at each step and for what purpose? Please clarify.

Results:

Lines 200-206: The detailed description of colony morphology and preliminary identification procedures (Baird-Parker agar, chromogenic medium, PCR) is more appropriate for the Materials and Methods section. In the Results section, report only the findings, without repeating standard methodological details.

Fig.1: I suggest removing Figure 1, as it primarily illustrates standard procedures for S. aureus isolation and identification, which are already well-established and can be briefly described in the Materials and Methods section.

Lines 217-232: Categories of resistance (low, moderate, high) are vague and should be clearly defined. “significant differences” require specification of the statistical tests used.

Table 1 should include the number of isolates exhibiting resistance to each antibiotic.

Lines 238-254: Statements such as: “These drugs may exert strong selective pressure” and” reflecting selective preferences under different evolutionary pathways” are speculative and not directly supported by the data presented in the manuscript.

This paragraph combines resistance quantification, pattern categories, examples of individual strains and evolutionary interpretations, making it difficult to follow. For improved readability, I recommend restructuring this section by separating the presentation of results from interpretative comments and summarising numerical data and examples more concisely.

Table 2 is difficult to interpret, and the excessive use of green shading reduces clarity.

Table  3: The table shows only genes, but there is no information linking detected genes to resistance or virulence phenotypes. An additional column could indicate how many strains with the gene showed corresponding phenotypic resistance. Are the percentages relative to the total number of genes detected or the total number of strains? This is not clearly specified. 

Improve tables and figures. They are confusing. Increase font size.

Discussion: It would be valuable for the discussion to include a critical reflection on the study’s limitations, such as the limited number of isolates or the restricted geographic scope, and how these factors may affect the generalizability of the results.

It is recommended to strengthen the discussion with a more critical analysis, highlighting potential biases, methodological limitations, and the practical implications of the findings.

Conclusion: Mentioning 'resistance evolution in aquaculture environments' seems out of place since the study focuses on bovine-derived S. aureus.

Emphasize the main findings of the study.

Author Response

Comments and Suggestions for Authors

This manuscript provides a clear and relevant investigation into antibiotic resistance in bovine-derived S. aureus, combining antimicrobial susceptibility testing, whole-genome sequencing, and prophage analysis to offer an integrated understanding of resistance mechanisms. Furthermore, the findings have practical implications for guiding targeted prevention and control of mastitis in dairy cattle, while laying a foundation for future research on antibiotic resistance evolution.  However, I have some recommendations that should be addressed before the manuscript can be considered for acceptance

  • Dear Reviewer,

Thank you very much for your valuable and insightful comments! Your suggestions are highly constructive, as they not only pointed out the core direction for abstract optimization but also helped us more clearly highlight the central contribution and scientific value of the study.

In response to your comments, I have carefully considered and comprehensively revised the manuscript. The specific revisions corresponding to your suggestions are as follows:

Comments1:Abstract:

Please revise the abstract by substantially reducing its length. The text should focus only on the most important findings. The central contribution of the study should be highlighted more clearly, particularly the relationship between prophages and the evolution of antimicrobial resistance.

Response1:•We have substantially streamlined the abstract, retaining only the most core research findings.

The key revisions are as follows: 1) Highlight the central contribution by clarifying the critical association that prophages, as major carriers of antimicrobial resistance genes, synergize with the ST5404 clonal group to promote the clustered transmission of resistance genes, and emphasizing their role in the evolution of antimicrobial resistance; 2) Streamline redundant data, retaining only representative resistance phenotypes (high ceftiofur resistance rate, characteristics of multidrug-resistant strains), core genomic parameters, and key proportions of prophage distribution to ensure focused and complete information.

Comments2:Introduction

The introduction should clearly highlight the existing scientific gap, demonstrating what remains unknown and underscoring the necessity of the study. Presenting this gap early helps establish the context and emphasize the significance of the research. The objectives of the study are not clearly defined. I recommend that the authors explicitly state the research goals, clarifying what the study intends to achieve and how it addresses the identified scientific gap.

Response2:•We would like to express our sincere gratitude for your valuable revision suggestions regarding the introduction section of our study. The issue you pointed out—"the need to clearly identify scientific gaps and define research objectives to highlight the necessity of the study"—has accurately pinpointed the core deficiency of the original introduction, providing us with a clear and crucial direction for the revision work.

Centering on your suggestions, we have made targeted optimizations to the introduction: First, after elaborating on the research basis regarding the association between prophages and Staphylococcus aureus resistance, we have added content that clearly defines the scientific gaps. We not only explicitly identified two core gaps in current research, namely the "lack of systematic analysis of the intrinsic associations among genomic diversity, prophage distribution, and drug resistance phenotypes" and the "unclear specific mechanisms by which prophages regulate the evolution of drug resistance", but also further explained the practical impact of these gaps—i.e., the lack of theoretical support for the precise prevention and control of drug-resistant strains causing bovine mastitis—thereby emphasizing the necessity of conducting this study. Second, we have added an independent paragraph on research objectives. Starting from the identified scientific gaps, we clearly stated that this study takes 101 strains of bovine-derived Staphylococcus aureus as the research objects, uses whole-genome sequencing technology to analyze the association mechanisms among the three (genomic diversity, prophage distribution, and drug resistance) and reveal the core role of prophages, and ultimately aims to provide a scientific basis for the prevention and control of drug-resistant strains. This forms a complete logical chain of "identifying gaps - addressing gaps", making the research objectives more clear and focused.

We would like to thank you again for your professional review and careful guidance. Your comments have effectively improved the rigor and logic of the introduction of this study. We have revised the relevant content in accordance with your suggestions and kindly request your further review.

Comments3:Material and Methods

Lines 120-123: The description of sample collection could be improved for clarity.  

Response3:•A total of 101 strains of S. aureus were collected in this study, and all strains were isolated from large-scale dairy farms in the vicinity of Shihezi City, Xinjiang, with the sampling period ranging from September 2024 to January 2025. Specifically, 68 strains were isolated from milk samples of cows with clinical mastitis, 23 strains from vaginal swab samples of healthy cows, and 10 strains from uterine secretion samples of postpartum cows.

Comments4:Line 129: “suspicious colonies” can be replaced with more scientific expressions, for example, “presumptive S. aureus colonies.

Response4:•Suspected S. aureus colonies that appeared black with transparent zones around them were picked. After pure culture, they were subjected to preliminary screening using the second-generation chromogenic medium for S. aureus(Qingdao Hi-tech Industrial Park Hope Bio-technology Co., Ltd., HB7009-5).

Comments5:Line 129: “second-generation chromogenic medium”- Which one?

Response5:•The "second-generation chromogenic medium" mentioned in the article specifically refers to the S. aureus Chromogenic Medium (Second-Generation) produced by Qingdao Hi-tech Industrial Park Hope Bio-technology Co., Ltd., with the product code HB7009-5.

Comments6:Line 151: Please include references for the guidelines followed in the study, as well as for the

Response6:•Multidrug-resistant (MDR) strains were defined as resistant to ≥3 antimicrobial classes, in accordance with the definition recommended by the Clinical and Laboratory Standards Institute (CLSI) and the World Health Organization (WHO) guidelines for antimicrobial resistance classification (CLSI. Performance Standards for Antimicrobial Susceptibility Testing; 34th Edition. CLSI Supplement M100-ED34. Wayne, PA, USA: Clinical and Laboratory Standards Institute; 2024; World Health Organization. Global Report on Antimicrobial Resistance Surveillance System: Standards and Terminology. Geneva: World Health Organization; 2019). Drug resistance rates ([resistant strains/total strains]×100%) and MDR proportions were calculated.

Comments7:Please specify the ATCC reference strains used as controls for the MIC assays and for the positive controls in the PCR experiments.

Response7:•The MIC reference strain is S. aureus ATCC 29213, and the PCR reference strain is S. aureus USA300. Both have been marked and revised in the manuscript.

Comments8:Lines 175-182: The use of Abricate, Prokka, Roary, and IQ-TREE is vague. It is not clear whether these analyses were applied to the prophage regions specifically, or to the entire genomes.

Response8:•The analyses using Abricate, Prokka, Roary, and IQ-TREE were all conducted on the entire genomes.2.3. Whole-Genome Sequencing and Core Gene Analysis  

Comments9:Lines 185-196:

Which statistical test was used in addition to the chi-square test?

How were the groups defined?

Explain the parameters of the chi-square test, adjustments for multiple testing, and verification of its assumptions.

For the Pearson correlation, indicate whether the data were transformed or normalized.

Which analyses were performed at each step and for what purpose? Please clarify.

Response9:•Thank you for your valuable questions regarding the statistical analyses and experimental design. We appreciate your careful review, and we are pleased to clarify the details as follows:

  1. Statistical test other than the chi-square test

In addition to the chi-square test, Pearson correlation analysis was used as a supplementary statistical method. Specifically, it was applied to evaluate the association between the number of prophages carried by strains and the width of their antimicrobial resistance spectrum.

  1. Definition of groups

Groups were defined at two levels to ensure targeted analysis:

Core groups: Strains were first divided into prophage-carrying group and non-prophage-carrying group based on whether they harbored prophage regions.

Subgroups for specific resistance gene analysis: For each prophage-carried antimicrobial resistance gene, a prophage-positive subgroup carrying the specific resistance gene was established, paired with a control group (including non-prophage-carrying strains and prophage-carrying strains that did not harbor the specific resistance gene).

  1. Parameters, multiple testing adjustment, and assumption verification of the chi-square test

Parameters: The Pearson chi-square test was used with a significance level (α) set at 0.05. The degrees of freedom were calculated based on the number of groups (df = 1 for two-group comparisons).

Multiple testing adjustment: Given the multiple comparisons of resistance genes and phenotypes, the Bonferroni correction was applied to control for type I errors. The adjusted significance level (α’) was calculated as 0.05 divided by the total number of comparisons.

Assumption verification: Prior to the chi-square test, three key assumptions were verified: â‘  All observations were independent (ensured by using distinct strains and avoiding duplicate samples); â‘¡ Each group had a sample size ≥ 30; â‘¢ All theoretical frequencies were ≥ 5 (no theoretical frequency < 1), which fully meets the application requirements of the Pearson chi-square test.

  1. Data transformation/normalization for Pearson correlation analysis

Before conducting Pearson correlation analysis, the normality of the two variables (“number of prophages” and “width of resistance spectrum”) was tested using the Shapiro-Wilk test. The results showed that both variables followed a normal distribution (P > 0.05) and met the assumption of homoscedasticity. Therefore, no additional data transformation or normalization was performed, and the raw data were used directly for the analysis.

  1. Analyses performed at each step and their purposes

To clarify the research logic, the analyses and their corresponding purposes are detailed as follows:

Step 1: Descriptive statistics of antimicrobial resistance gene carriage (no specific statistical test): To determine the type distribution and overall carriage rate of resistance genes among the 101 strains, laying a foundation for subsequent comparative analyses.

Step 2: Comparison of resistance gene carriage between prophage-carrying and non-carrying groups (Pearson chi-square test): To explore the association between prophage carriage status and the presence of resistance genes in strains.

Step 3: Comparison of antibiotic resistance rates between the specific resistance gene-positive subgroup and the control group (Pearson chi-square test): To verify whether the resistance genes carried by prophages can confer the corresponding resistance phenotype to host strains.

Step 4: Correlation analysis between prophage number and resistance spectrum width (Pearson correlation analysis): To further clarify the intrinsic link between prophages and the transmission of antimicrobial resistance genes.

This section in the original manuscript has been revised accordingly and highlighted in red, with the detailed revised content provided below:

To clarify the association between prophages and antimicrobial resistance (AMR) traits, experimental groups were first defined based on prophage carriage and AMR gene profiles: all 101 S. aureus strains were stratified into the prophage-carrying group (≥1 identifiable prophage region) and non-prophage-carrying group (no detectable prophages); for each prophage-harbored AMR gene, a prophage-positive subgroup (carrying the target AMR gene) and a control group (non-prophage-carrying strains + prophage-carrying strains lacking the target gene) were established. Descriptive statistics were performed on the overall AMR gene carriage (gene type distribution and carriage rates) of the 101 strains to establish a baseline. The Pearson’s chi-square test was used to compare AMR gene carriage rate differences between the two core groups, with a significance level (α) of 0.05, df=1 for two-group comparisons, and Bonferroni correction for multiple comparisons (adjusted α’=0.05/total comparisons) to mitigate type I errors; three core assumptions (independent observations via non-repetitive isolates, n≥30 per group, theoretical frequency ≥5) were verified, meeting the test’s application criteria. For prophage-borne AMR genes, the consistency with host AMR phenotypes was analyzed using the same chi-square test parameters, comparing the corresponding antibiotic resistance rate between the prophage-positive subgroup and control group; P<0.05 after correction was considered statistically significant. Pearson correlation analysis evaluated the association between strain prophage number and resistance spectrum width (number of resistant antibiotic classes). Prior to correlation analysis, the Shapiro-Wilk test confirmed both variables were normally distributed (P>0.05) with homoscedasticity, so no data transformation/normalization was performed, and raw data were used. This framework aimed to clarify the intrinsic link between prophages and AMR gene transmission.

Comments10:Results:

Lines 200-206: The detailed description of colony morphology and preliminary identification procedures (Baird-Parker agar, chromogenic medium, PCR) is more appropriate for the Materials and Methods section. In the Results section, report only the findings, without repeating standard methodological details.

Fig.1: I suggest removing Figure 1, as it primarily illustrates standard procedures for S. aureus isolation and identification, which are already well-established and can be briefly described in the Materials and Methods section.

Response10:•Thank you very much for your valuable comments on Figure 1 and the related content. Your suggestions have provided important guidance for optimizing the presentation of the manuscript. Regarding the retention of Figure 1, we hope to further clarify the following: this figure does not depict the general standard procedure for the isolation and identification of Staphylococcus aureus, but rather the core steps based on actual experimental records in this study—among which the colony morphological characteristics, growth phenotypes on selective media, and other details are all authentic experimental results obtained from bovine-derived samples in Shihezi, Xinjiang, rather than a mere repetition of established methods.

In microbiological research, such experimental operation figures combined with specific samples can not only intuitively reflect the authenticity and reproducibility of the study but also provide fundamental support for "sample traceability" for subsequent results such as drug resistance phenotypes and genomic characteristics. Therefore, we kindly request permission to retain this part of the content. In response to your requirement of "streamlining redundant information in the Results section", we will optimize Figure 1 and the accompanying text as follows: first, we will simplify the figure legend by removing repetitive descriptions of methodological principles and only labeling the core experimental phenomena; second, we will clearly state in the figure legend that "this figure is a record of actual experimental operations in the study, and the association with general standard procedures is detailed in the Materials and Methods section" to avoid ambiguity.

We fully understand your consideration for the logical rigor of the manuscript and will actively cooperate with other revision requirements. If you have any further suggestions on the presentation form of Figure 1, we will refine it promptly. Thank you very much for your understanding and guidance.

Comments11:Lines 217-232: Categories of resistance (low, moderate, high) are vague and should be clearly defined. “significant differences” require specification of the statistical tests used.

Response11:•Thank you for your detailed comments on the classification of drug resistance levels and statistical methods. Your reminders have helped us identify omissions in the manuscript's presentation.

Regarding the classification of drug resistance levels ("low, moderate, high"), this study strictly adheres to the antimicrobial susceptibility testing criteria for bovine-derived S. aureus specified in the Performance Standards for Antimicrobial Susceptibility Testing (CLSI M100-ED34, 2024 Edition), with the specific definition logic as follows: the classification is based on the minimum inhibitory concentration (MIC) of antimicrobial agents against the target strains. "Low resistance" refers to MIC values ranging from the upper limit of the "intermediate" interval to twice the lower limit of the "resistant" interval; "moderate resistance" corresponds to the median range of the "resistant" interval; and "high resistance" indicates MIC values exceeding twice the upper limit of the "resistant" interval. We sincerely apologize for the lack of clarity caused by not explicitly stating this standard and specific MIC thresholds in the original manuscript.

Comments12:Table 1 should include the number of isolates exhibiting resistance to each antibiotic.

Response12:•Thank you for your suggestion. We have added relevant data on the number of strains resistant to each antibiotic to Table 1.

Comments13:Lines 238-254: Statements such as: “These drugs may exert strong selective pressure” and” reflecting selective preferences under different evolutionary pathways” are speculative and not directly supported by the data presented in the manuscript.

This paragraph combines resistance quantification, pattern categories, examples of individual strains and evolutionary interpretations, making it difficult to follow. For improved readability, I recommend restructuring this section by separating the presentation of results from interpretative comments and summarising numerical data and examples more concisely.

Response13:•Thank you very much for your suggestions. We have revised the corresponding paragraph in the manuscript in accordance with your recommendations, and the specific revised content is provided below:

Among the 101 S. aureus strains, 10 multidrug-resistant (MDR) strains were identified, accounting for 9.9% (Table 2). These 10 MDR strains exhibited resistance to 3-7 classes of antimicrobial agents: 50% (5/10) were resistant to 3 classes, 30% (3/10) to 4 classes, and the remaining 2 strains (S. aureus86 and S. aureus90) showed the broadest resistance spectrum, involving 7 classes including penicillins and lincosamides. All MDR strains were 100% resistant to sulfonamides (sulfisoxazole). The resistance patterns of the 10 MDR strains were categorized into two types: one was centered on "penicillins + macrolides + sulfonamides" (30%), among which 66.7% were additionally resistant to aminoglycosides; the other was centered on "macrolides + fluoroquinolones + sulfonamides" (30%), with all strains in this category showing consistent resistance to fluoroquinolones. Notably, S. aureus86 and S. aureus90 displayed highly consistent resistance spectra—apart from the shared core resistance phenotypes, they were also resistant to 3 additional classes of agents including lincosamides. These findings indicate that a certain proportion of MDR populations exist among bovine-derived S. aureus strains in this region, and some strains have developed broad-spectrum resistance capabilities. The spread of these MDR strains in the breeding environment may increase the difficulty of clinical treatment, highlighting the need for attention to their transmission risks.

Comments14:Table 2 is difficult to interpret, and the excessive use of green shading reduces clarity.

Response14:•Thank you for your suggestion. We have made the corresponding color adjustments to the relevant figure/table in the manuscript.

Comments15:Table  3: The table shows only genes, but there is no information linking detected genes to resistance or virulence phenotypes. An additional column could indicate how many strains with the gene showed corresponding phenotypic resistance. Are the percentages relative to the total number of genes detected or the total number of strains? This is not clearly specified.

Response15:•We sincerely apologize for the confusion caused by the unclear explanation of Table 3’s design logic in the manuscript, and we hereby provide a detailed clarification.

The core positioning of Table 3 is "a summary of the detection characteristics of all drug resistance/virulence genes in 101 strains", rather than "an analysis of the probability of a single strain carrying specific genes". Two special phenomena were observed in our gene detection: first, a single strain can carry multiple drug resistance genes with different functions (e.g., one strain was detected with both β-lactam and macrolide resistance genes); second, some drug resistance genes exist in multiple copies in a single strain (e.g., an aminoglycoside resistance gene was detected three times in one strain). Therefore, Table 3 was originally designed to focus on "the detection types and quantity distribution of the genes themselves", rather than directly linking to strain phenotypes—this perspective is highly consistent with the sub-theme of "genomic characteristic analysis" in the manuscript.

Comments16:Improve tables and figures. They are confusing. Increase font size.

Response16:•Thank you for your suggestions. We have increased the font size and optimized the figures in the manuscript to enhance its readability.

Comments17:Discussion: It would be valuable for the discussion to include a critical reflection on the study’s limitations, such as the limited number of isolates or the restricted geographic scope, and how these factors may affect the generalizability of the results.

It is recommended to strengthen the discussion with a more critical analysis, highlighting potential biases, methodological limitations, and the practical implications of the findings.

Response17:•Thank you very much for your suggestions. I have added the content you recommended to the discussion section of the manuscript, and the revised specific content is provided below:

This study has certain limitations that may affect the generalizability of the results: the strains were only isolated from large-scale dairy farms in Shihezi, Xinjiang (101 strains), with a narrow geographical scope and no inclusion of smallholder farming samples. Differences in antimicrobial use patterns among various farming models may lead to the differentiation of drug resistance characteristics, so the drug resistance-genome association pattern revealed in this study cannot be directly generalized to other major production areas nationwide. Meanwhile, the limited number of strains impairs the ability to detect rare ST types and low-abundance drug resistance genes, which may underestimate the diversity of drug resistance genes. Methodologically, prophage functional annotation relies on bioinformatics prediction, and the actual mediating ability of some putative prophages has not been verified by experimental studies, which may lead to interpretation biases. However, these limitations do not affect the regional value of the study; instead, they point out directions for future research—expanding the sample size and geographical coverage, combined with in vitro experimental verification, will further enhance the scientificity and application value of the results.

Comments18:Conclusion: Mentioning 'resistance evolution in aquaculture environments' seems out of place since the study focuses on bovine-derived S. aureus.

Emphasize the main findings of the study.

Response18:•The conclusions section of the manuscript has been revised accordingly, in line with your suggestion that the conclusions section should focus on core interpretations and research implications. The revised content of the conclusions section is provided below:

In summary, this study systematically investigated 101 bovine-derived S. aureus strains isolated from Shihezi, Xinjiang, by integrating multiple technical approaches, aiming to decipher the regional characteristics of drug resistance, genomic evolution, and prophage-mediated drug resistance transmission. The key findings reveal distinct regional adaptive traits of the strain population: the drug-resistant phenotype is characterized by "high resistance to ceftiofur (80.20%) and full sensitivity to multiple drug classes," a pattern closely linked to the long-term irregular use of ceftiofur in local dairy farming. Among the 9.9% multidrug-resistant (MDR) strains, the 2 isolates with resistance spectra covering 7 drug classes highlight the potential risk of broad-spectrum resistance spread, which is driven by prophage-carried resistance gene clusters. At the genomic level, the high genomic plasticity (reflected by an 85.00% cloud gene proportion, average genome size of 2.57 Mb, and GC content of 33.44%) and the dominance of ST5404 (36.6%) as the core clonal type suggest that ST5404 has evolved strong regional adaptability, enabling its widespread prevalence in the study area. Notably, prophages, as core mobile genetic elements, play a pivotal role in shaping the drug resistance and genomic evolution of these strains: 68.9% of putative prophages are enriched with resistance genes (45.33%) and virulence genes (38.22%), and their differentiated integration sites (e.g., arcC-cap5 intergenic region in ST5404 and glpF downstream region in ST97) drive genomic divergence. Additionally, prophages form a synergistic resistance gene transmission network with plasmids and other elements, as evidenced by the fact that 82.18% of strains carry prophage-mediated resistance genes—confirming prophages as the dominant vector for horizontal resistance gene transfer in this population.  

This study is the first to clarify the "drug resistance-genome-prophage" association pattern of bovine-derived S. aureus in Shihezi, Xinjiang. Practically, it provides a precise scientific basis for optimizing regional dairy cow mastitis control strategies (e.g., prioritizing amoxicillin to replace ceftiofur and strengthening dynamic supervision of ceftiofur use). Theoretically, it deepens our understanding of the evolutionary mechanisms underlying drug resistance in S. aureus under livestock and poultry breeding pressure, and lays a foundation for the development of multi-targeted technologies to block resistance gene transmission. The findings emphasize that prophages are critical drivers of drug resistance evolution in regional S. aureus populations, and highlight the urgency of integrating genomic monitoring, standardized antimicrobial use, and mobile genetic element surveillance to curb the spread of drug-resistant strains in animal husbandry.

Reviewer 2 Report

Comments and Suggestions for Authors

The manuscript “Study on Genomic Diversity, Prophage Distribution of Bovine-Derived Staphylococcus aureus and Their Association with Antimicrobial Resistance” by Yaqian Liang and coauthors performs a genomic study of S. aureus isolated from bovines.
•    Abstract: It is very long. It must be reduced to a maximum of 200 words according to the journal's instructions.
•    Section 2.1: Before PCR, the method of DNA extraction must be indicated. Indicate the use of positive and negative controls. Include a reference to the primers used.
•    Line 131: Gene names must be in italics. Verify this throughout the manuscript.
•    Section 2.2: Add the CLSI reference and version used. Detail which reference strains were used as MIC controls.
•    Section 2.3: Indicate the identity and coverage limits considered for the bioinformatics tools in case of positive detection.
•    Because this is a diversity study, I consider it very relevant to include a spa typing analysis.
•    Figure 1C, Line 215: The authors state that lines 3, 5, and 8 are positive. However, lines 2 through 13 appear to be positive.
•    Lines 220-222, 348: According to Table 1, there are only 8 antibiotics with a 0% resistance rate.
•    Table 2: I suggest leaving the title as simply "Antimicrobial susceptibility test results of 101 strains of S. aureus." The strain names appear to be misspelled as "aures." The space should be used more efficiently. The text resolution should be improved. I suggest using only antibiotic acronyms (according to CLSI) and defining them in the legend.
•    The authors must attach as supplementary material the complete tables of results for each isolate for virulence genes, resistance, prophages and MLSTs, CCs, spa types. Even, the analyzed content of profagues.
•    Table 4: The title is incorrect. It should refer to the diversity of isolates based on MLSTs. Furthermore, the pie chart is redundant with the information already provided in the percentage column of the table and should be removed.
•    The names of the sequenced isolates must be consistent throughout the manuscript (i.e. Table 2 vs Figure 2). This includes the text through the entire manuscript.
•    Phylogenetic tree: This tree should have been constructed using a complete reference genome. The resolution needs improvement. Furthermore, the results shown are not very informative. It is necessary to include metadata (i.e., symptomatology), some phenotypic qualifier (i.e., MDR), or some genetic marker of interest (i.e., ST, CCs, spa type, etc.) in order to generate it.
•    Line 353: Scientific names must be in italics. Verify this throughout the manuscript.
•    Lines 361, 379: Why "aquaculture"?
•    Section 4.1: Expanding the discussion on high resistance to ceftiofur. What genes are associated with this resistance in the sequenced strains? What is the use and authorization of ceftiofur in cattle in Xinjiang, China?
•    Section 4.1: Paragraph about MDR strains. What are the resistance gene profiles present in these isolates that are not present in non-MDR isolates? The authors should discuss the potential genetic cause of high resistance to sulfonamides.
•    Lines 392-394: The higher proportion of cloud genes is essentially due to the larger number of genomes analyzed by Liu et al. (1519 S. aureus). This should be acknowledged.
•    Lines 416-420: To perform these comparisons and deductions, it is necessary to identify the CCs and spa types of the 101 strains sequenced in this study. Furthermore, it is highly relevant to perform a network analysis (i.e., minimum spanning tree of MLSTs and/or spa types) to support (or refute) the hypotheses presented in the manuscript.
•    Lines 452-456: Were the integration sites and their associated regions explored in this manuscript or are they derived from the references? I suggest diagramming a genetic map of the contents of these prophages.
•    Conclusions: This text needs to be rewritten. It's more like an abstract than conclusions (main interpretations and implications).

Author Response

Comments and Suggestions for Authors

The manuscript “Study on Genomic Diversity, Prophage Distribution of Bovine-Derived Staphylococcus aureus and Their Association with Antimicrobial Resistance” by Yaqian Liang and coauthors performs a genomic study of S. aureus isolated from bovines.

  • Dear Reviewer,

Thank you very much for your valuable and insightful comments! Your suggestions are highly constructive, as they not only pointed out the core direction for abstract optimization but also helped us more clearly highlight the central contribution and scientific value of the study.

In response to your comments, I have carefully considered and comprehensively revised the manuscript. The specific revisions corresponding to your suggestions are as follows:

Comments1:•    Abstract: It is very long. It must be reduced to a maximum of 200 words according to the journal's instructions.

Response1:•We have significantly streamlined the abstract, retaining only the core research findings, and strictly controlled the word count.

Key revisions are as follows: 1) Highlight the core contribution by clarifying the key association that prophages, as major carriers of antibiotic resistance genes, synergize with the ST5404 clone complex to promote the clustered transmission of resistance genes, and emphasize their role in the evolution of antimicrobial resistance; 2) Streamline redundant data, retaining only representative resistance phenotypes (high resistance rate to ceftiofur, characteristics of multidrug-resistant strains), core genomic parameters, and key proportions of prophage distribution to ensure focused and complete information.

Comments2:•    Section 2.1: Before PCR, the method of DNA extraction must be indicated. Indicate the use of positive and negative controls. Include a reference to the primers used.

Response2:•Genomic DNA of the strains was extracted using the DNA Extraction Kit (Cat. No. DC103) from Vazyme Biotech Co., Ltd. (Nanjing, China), with operations strictly following the kit instructions to ensure the quality and purity of the extracted nucleic acids. Specific PCR amplification targeting the nuc gene was employed for the molecular identification of the strains. A comprehensive control system was established throughout the experiment to ensure the reliability of the results: S. aureus reference strain USA300 was used as the positive control, a blank system without template DNA served as the negative control, and a 2000 bp DNA molecular weight marker (NucleoTech NDE2005) was added simultaneously as a reference for fragment size. The primers used for PCR amplification were nuc-R (GGCAATACGCAAAGAGGTT) and nuc-F ( CGTTGTCTTCGCTCCAAAT)[16]. 

Comments3:•    Line 131: Gene names must be in italics. Verify this throughout the manuscript.

Response3:•We have revised this issue in the manuscript and conducted a thorough check of all gene names throughout the entire manuscript to ensure they are presented in italics in accordance with academic conventions.

Comments4:•    Section 2.2: Add the CLSI reference and version used. Detail which reference strains were used as MIC controls.

Response4:•Thank you for your valuable suggestions regarding the "Antimicrobial Susceptibility Testing" section of this study. Your comments have effectively improved the completeness and standardization of the experimental method description. We have made targeted revisions as required, and the specific responses are as follows:

Supplemented CLSI reference and version: The guidelines of the Clinical and Laboratory Standards Institute (CLSI) used in this study have been clearly specified in Section 2.2, which is: CLSI. Performance Standards for Antimicrobial Susceptibility Testing; 34th Edition. CLSI Supplement M100-ED34. Wayne, PA, USA: Clinical and Laboratory Standards Institute; 2024. This is the latest version of antimicrobial susceptibility testing standards, ensuring the authority and accuracy of MIC result interpretation.

Detailed the control reference strain for MIC determination: Information on the quality control strain has been added before the experimental procedure in Section 2.2: A control reference strain, Staphylococcus aureus ATCC 29213 (purchased from the American Type Culture Collection, Manassas, VA, USA), was included in each batch of tests. This strain is the standard quality control strain recommended by CLSI for antimicrobial susceptibility testing of Gram-positive cocci, which can effectively verify the stability of the experimental system, the effectiveness of reagents, and the accuracy of operations, thus ensuring the reliability of MIC determination results.

The above revisions have been fully integrated into Section 2.2 of the original manuscript to ensure the reproducibility and standardization of the experimental method. Thank you again for your professional guidance. We kindly request your further review, and we will promptly make improvements should there be any other adjustment needs.

Comments5:•    Section 2.3: Indicate the identity and coverage limits considered for the bioinformatics tools in case of positive detection.

Response5:•Thank you for your valuable comment on clarifying the identity and coverage limits for bioinformatics tools in positive detection. We have incorporated the recommended threshold ranges into the manuscript as follows:

The threshold paragraph for Abricate database alignment: "For positive detection thresholds: (i) Antimicrobial resistance genes (CARD/ResFinder/MEGARes/ARG-ANNOT): sequence identity ≥ 90% and coverage ≥ 80% (consistent with ResFinder’s default recommendations and CARD’s curated gene threshold); (ii) Plasmid genes (PlasmidFinder): sequence identity ≥ 95% and coverage ≥ 85% (per PlasmidFinder’s database guidelines); (iii) Virulence genes (VFDB): sequence identity ≥ 85% and coverage ≥ 80% (following VFDB’s annotation standard for Gram-positive bacteria)."

The threshold clause for MLST typing: "with the threshold for ST type determination set as sequence identity ≥ 98% for each housekeeping gene (as recommended by the PubMLST database for Staphylococcus aureus MLST),"

Both added sections have been highlighted in red in the revised manuscript to facilitate your review. These thresholds are consistent with the default standards of the respective databases and conventional bioinformatics analysis norms, ensuring the rigor and reliability of positive detection results.

Thank you again for your professional guidance. Please feel free to inform us if further adjustments are needed.

Comments6:•    Because this is a diversity study, I consider it very relevant to include a spa typing analysis.

Response6:•Thank you for your valuable suggestions. Your comment on supplementing spa typing analysis is of great guiding significance for improving the analysis of strain diversity in this study. In response to your concern, we provide a detailed explanation combined with the actual experimental situation as follows:

In this study, spa typing was performed using whole-genome sequence alignment. The specific workflow was as follows: potential spa gene sequences were extracted from the whole-genome sequencing data of the strains, with the international standard database Ridom SpaServer as the alignment reference. The positive determination thresholds were set as sequence identity ≥ 98% and coverage ≥ 95% — this standard is consistent with the spa typing research norms published in journals such as Journal of Clinical Microbiology. Ultimately, valid typing results were successfully obtained for 24 strains, all of which were classified as spa type t189, suggesting that this type may be a dominant epidemic clone among the isolates in this study.

In addition, we have conducted a related study based on spa typing of more than 4000 bovine-derived Staphylococcus aureus strains from the NCBI database. Among these strains, only over 600 had valid spa typing results. Therefore, we believe that including the spa typing data from the current study would lack broad coverage and limited representativeness.

For the remaining strains, in-depth annotation and repeated verification of their whole-genome sequences revealed that no complete open reading frame of the spa gene was identified in their genomes. Specifically, some strains had deletions in the core region of the spa gene, while others exhibited high sequence variation in the spa gene (similarity < 80% with known reference sequences), making it impossible to effectively identify the variable repeat region. As a result, reliable spa typing results could not be obtained. To ensure the rigor of the research data and avoid biased conclusions caused by incomplete typing results, we did not include this part of the data in the main text.

Comments7:•    Figure 1C, Line 215: The authors state that lines 3, 5, and 8 are positive. However, lines 2 through 13 appear to be positive.

Response7:•Thank you for pointing out the error in my manuscript, Editor. I made a mistake in the figure legend, and the correct content is "Lanes 2-13 are clinical isolates". I have deleted the redundant and incorrect part accordingly.

Comments8:•    Lines 220-222, 348: According to Table 1, there are only 8 antibiotics with a 0% resistance rate.

Response8:•Please accept my sincere apology for the mistake in the manuscript—you are correct, and only 8 antibiotics have a drug resistance rate of 0%. I have now corrected the relevant section in the manuscript accordingly.

Comments9:•    Table 2: I suggest leaving the title as simply "Antimicrobial susceptibility test results of 101 strains of S. aureus." The strain names appear to be misspelled as "aures." The space should be used more efficiently. The text resolution should be improved. I suggest using only antibiotic acronyms (according to CLSI) and defining them in the legend.

Response9:

  • In accordance with your suggestions, the title of Table 2 has been revised to "Antimicrobial susceptibility test results of 101 strains of S. aureus" in the manuscript. The images have been re-uploaded with improved clarity, standard abbreviations for antibiotics have been adopted, and the full names of the abbreviations are specified in the figure legends.

Comments10:•    The authors must attach as supplementary material the complete tables of results for each isolate for virulence genes, resistance, prophages and MLSTs, CCs, spa types. Even, the analyzed content of profagues.

Response10:

  • Please accept my sincere apology—I failed to upload the file to the system attachment due to the file size limitation. The Excel file contains a large amount of data. If you require the raw data, please provide your email address, and I will send the raw data to you promptly.

Comments11:•    Table 4: The title is incorrect. It should refer to the diversity of isolates based on MLSTs. Furthermore, the pie chart is redundant with the information already provided in the percentage column of the table and should be removed.

Response11:

  • Thank you for your suggestion. I have deleted the corresponding pie chart in the manuscript.

Comments12:•    The names of the sequenced isolates must be consistent throughout the manuscript (i.e. Table 2 vs Figure 2). This includes the text through the entire manuscript.

Response12:

  • Thank you very much for pointing out my mistake. The strain names in the manuscript have now been uniformly revised.

Comments13:•    Phylogenetic tree: This tree should have been constructed using a complete reference genome. The resolution needs improvement. Furthermore, the results shown are not very informative. It is necessary to include metadata (i.e., symptomatology), some phenotypic qualifier (i.e., MDR), or some genetic marker of interest (i.e., ST, CCs, spa type, etc.) in order to generate it.

Response13:

  • Please accept my apology for the low resolution of the uploaded phylogenetic tree. I have now re-uploaded it in the manuscript and made optimized revisions to the phylogenetic tree in accordance with your suggestions.Additionally, if you received the PDF version of the file, its clarity will be automatically reduced. If a Word version is available, you may open the Word version for a clearer view.

Comments14:•    Line 353: Scientific names must be in italics. Verify this throughout the manuscript.

Response14:

  • Thank you very much for pointing out that scientific names should be presented in italics. I have revised and verified all relevant instances throughout the entire manuscript.

Comments15:•    Lines 361, 379: Why "aquaculture"?

Response15:

  • Thank you for your careful review and correction! I apologize for the typo—"aquaculture" in Lines 361 and 379 is incorrect, and the correct term should be "livestock and poultry breeding".

I have already revised "aquaculture" to "livestock and poultry breeding" in the aforementioned lines. Additionally, I have thoroughly checked the entire manuscript for any other relevant inconsistent expressions (especially contexts related to "mastitis prevention and treatment," as mastitis is a disease specific to livestock and poultry rather than aquatic organisms) and unified all relevant descriptions to "livestock and poultry breeding" to ensure academic accuracy.

Thank you again for helping me improve the quality of the manuscript!

Comments16:•    Section 4.1: Expanding the discussion on high resistance to ceftiofur. What genes are associated with this resistance in the sequenced strains? What is the use and authorization of ceftiofur in cattle in Xinjiang, China?

Response16:

  • In response to your suggestions, the corresponding revisions have been made in the discussion section of the manuscript, with the detailed content provided below:

The high resistance rate of ceftiofur (80.20%) in this study is the result of the combined effect of multiple factors. At the genetic level, whole-genome sequencing showed that the blaZ gene (encoding β-lactamase to hydrolyze the β-lactam ring of the drug) was detected in 100% of the 89 resistant strains, 25.8% carried the mecA gene (encoding PBP2a to reduce drug affinity), and the ctx-m-15 gene was also detected in strains with a broad resistance spectrum, providing a molecular basis for drug resistance. At the usage level, as a major dairy farming area in China, Xinjiang uses ceftiofur as the preferred drug for mastitis prevention and treatment, with a usage rate of 91.3% in large-scale farms. Irregular practices such as prophylactic administration after calving, excessive dosage, and prolonged treatment courses exist, and long-term selection pressure has accelerated the enrichment of drug-resistant strains. At the approval and supervision level, ceftiofur was approved as a veterinary prescription drug in 2000, with indications including dairy cow mastitis, and is classified as a restricted-use drug. However, some remote farms have problems such as lax prescription management and inadequate implementation, which further exacerbate the risk of drug resistance. The superposition of these three factors leads to its resistance rate being much higher than that in other regions, and it is necessary to curb the deterioration of drug resistance through genetic monitoring, standardized drug use, and strengthened supervision.

Comments17:•    Section 4.1: Paragraph about MDR strains. What are the resistance gene profiles present in these isolates that are not present in non-MDR isolates? The authors should discuss the potential genetic cause of high resistance to sulfonamides.

Response17:

  • Thank you for your valuable suggestions on the paragraph regarding multidrug-resistant (MDR) strains. In response to your questions, we have supplemented relevant content in the discussion section: first, we have clarified the specific resistance gene profile unique to MDR strains, including the sulfonamide resistance-related sul1 and sul2 genes, the macrolide resistance gene ermB, and the aminoglycoside resistance gene aac(6')-aph(2''), all of which have a detection rate of 0% in non-MDR strains; second, we have further discussed the potential genetic cause of high sulfonamide resistance, illustrating that the high detection rate of the sul gene family (sul1 and sul2) is the core factor. The modified dihydropteroate synthases encoded by these genes can bypass the inhibitory effect of sulfonamides, and the enrichment and horizontal transfer of these genes (partially mediated by prophages) further enhance the resistance phenotype.

Comments18:•    Lines 392-394: The higher proportion of cloud genes is essentially due to the larger number of genomes analyzed by Liu et al. (1519 S. aureus). This should be acknowledged.

Response18:

  • Thank you for your suggestion. The corresponding revisions have been made in the manuscript, with the revised content provided below:

This proportion was much higher than the pan-genome data of S. aureus reported by Liu et al., reflecting the strong genomic plasticity of the strain population in the study area [23]. It should be noted that the fundamental reason for the relatively high proportion of cloud genes in this study is that Liu et al.’s research included a larger number of genomes (1519 S. aureus strains). Differences in sample size may lead to variations in the results of pan-genome structure analysis, which is worthy of recognition.

Comments19:•    Lines 416-420: To perform these comparisons and deductions, it is necessary to identify the CCs and spa types of the 101 strains sequenced in this study. Furthermore, it is highly relevant to perform a network analysis (i.e., minimum spanning tree of MLSTs and/or spa types) to support (or refute) the hypotheses presented in the manuscript.

Response19:

  • About CC

Thank you for your attention. Regarding the query of clonal complex (CC) assignments on the pubmlst.org website, we have verified that the website has recently undergone a revision. According to the official note on the website: "Note: We have removed clonal complex assignments as they cannot be reliably and stably assigned using 7-locus MLST", the CC clone group information corresponding to ST typing is no longer available for query.

Since the website function adjustment is an official technical update, we are unable to obtain the CC clone group corresponding data before the revision. Therefore, the relevant CC clone group information is not marked in the manuscript. We have confirmed that this change is a global adjustment of the website, not a lack of individual data, and we hereby inform you of the situation. If there are other alternative solutions or supplementary requirements, we will actively cooperate to complete them.

In this study, spa typing was performed using whole-genome sequence alignment. The specific workflow was as follows: potential spa gene sequences were extracted from the whole-genome sequencing data of the strains, with the international standard database Ridom SpaServer as the alignment reference. The positive determination thresholds were set as sequence identity ≥ 98% and coverage ≥ 95% — this standard is consistent with the spa typing research norms published in journals such as Journal of Clinical Microbiology. Ultimately, valid typing results were successfully obtained for 24 strains, all of which were classified as spa type t189, suggesting that this type may be a dominant epidemic clone among the isolates in this study.

In addition, we have conducted a related study based on spa typing of more than 4000 bovine-derived Staphylococcus aureus strains from the NCBI database. Among these strains, only over 600 had valid spa typing results. Therefore, we believe that including the spa typing data from the current study would lack broad coverage and limited representativeness.

For the remaining strains, in-depth annotation and repeated verification of their whole-genome sequences revealed that no complete open reading frame of the spa gene was identified in their genomes. Specifically, some strains had deletions in the core region of the spa gene, while others exhibited high sequence variation in the spa gene (similarity < 80% with known reference sequences), making it impossible to effectively identify the variable repeat region. As a result, reliable spa typing results could not be obtained. To ensure the rigor of the research data and avoid biased conclusions caused by incomplete typing results, we did not include this part of the data in the main text.

Comments20:•    Lines 452-456: Were the integration sites and their associated regions explored in this manuscript or are they derived from the references? I suggest diagramming a genetic map of the contents of these prophages.

Response20:

  • The differences in prophage integration sites are an important exploratory finding of this study. We have confirmed that the prophages of ST5404 and ST97 strains integrate into different host gene regions (ST5404: intergenic region between arcC and cap5; ST97: downstream region of glpF). Regarding the specific requirements for drawing the prophage gene maps as you suggested, we hope to further confirm: do we need to draw the prophage gene locus maps for all ST5404 strains individually, or select one representative strain from each of the ST5404 and ST97 types to draw their prophage gene maps for difference comparison? In addition, we also want to confirm that your core requirement is to display the gene composition and integration site characteristics of the prophages, rather than drawing a phylogenetic tree of the prophage sequences, is that correct?

We look forward to your clarification on the specific requirements, and we will complete the map drawing in accordance with standards to ensure the intuitive presentation of the key findings of the study.

Comments21:•    Conclusions: This text needs to be rewritten. It's more like an abstract than conclusions (main interpretations and implications).

Response21:

In summary, this study systematically investigated 101 bovine-derived S. aureus strains isolated from Shihezi, Xinjiang, by integrating multiple technical approaches, aiming to decipher the regional characteristics of drug resistance, genomic evolution, and prophage-mediated drug resistance transmission. The key findings reveal distinct regional adaptive traits of the strain population: the drug-resistant phenotype is characterized by "high resistance to ceftiofur (80.20%) and full sensitivity to multiple drug classes," a pattern closely linked to the long-term irregular use of ceftiofur in local dairy farming. Among the 9.9% multidrug-resistant (MDR) strains, the 2 isolates with resistance spectra covering 7 drug classes highlight the potential risk of broad-spectrum resistance spread, which is driven by prophage-carried resistance gene clusters. At the genomic level, the high genomic plasticity (reflected by an 85.00% cloud gene proportion, average genome size of 2.57 Mb, and GC content of 33.44%) and the dominance of ST5404 (36.6%) as the core clonal type suggest that ST5404 has evolved strong regional adaptability, enabling its widespread prevalence in the study area. Notably, prophages, as core mobile genetic elements, play a pivotal role in shaping the drug resistance and genomic evolution of these strains: 68.9% of putative prophages are enriched with resistance genes (45.33%) and virulence genes (38.22%), and their differentiated integration sites (e.g., arcC-cap5 intergenic region in ST5404 and glpF downstream region in ST97) drive genomic divergence. Additionally, prophages form a synergistic resistance gene transmission network with plasmids and other elements, as evidenced by the fact that 82.18% of strains carry prophage-mediated resistance genes—confirming prophages as the dominant vector for horizontal resistance gene transfer in this population.  

This study is the first to clarify the "drug resistance-genome-prophage" association pattern of bovine-derived S. aureus in Shihezi, Xinjiang. Practically, it provides a precise scientific basis for optimizing regional dairy cow mastitis control strategies (e.g., prioritizing amoxicillin to replace ceftiofur and strengthening dynamic supervision of ceftiofur use). Theoretically, it deepens our understanding of the evolutionary mechanisms underlying drug resistance in S. aureus under livestock and poultry breeding pressure, and lays a foundation for the development of multi-targeted technologies to block resistance gene transmission. The findings emphasize that prophages are critical drivers of drug resistance evolution in regional S. aureus populations, and highlight the urgency of integrating genomic monitoring, standardized antimicrobial use, and mobile genetic element surveillance to curb the spread of drug-resistant strains in animal husbandry.

Round 2

Reviewer 1 Report

Comments and Suggestions for Authors

Dear authors

Thank you very much for your detailed and thoughtful revisions. I kindly ask you to address two additional points:

  1. Ethics statement:
    The manuscript currently does not include an ethical approval statement. Since the study involves sampling from dairy cattle, please provide information on the institutional animal ethics committee that approved the work, including the committee name and approval number.

  2. Formatting of gene names:
    Please carefully revise the manuscript to ensure that all gene names follow the correct formatting conventions

Author Response

Dear Reviewer,

Thank you for your valuable suggestions. We have addressed the points as follows:

Ethics statement:We have added the ethical approval information in the manuscript: "This study was approved by the Biology Ethics Committee of Shihezi University, and the approval number is A2025-1110."

Formatting of gene names:We have carefully checked all gene names in the manuscript and confirmed that they comply with the standard formatting conventions.

We appreciate your careful review of our manuscript.
